
# 1 Arctic tropospheric ozone: assessment of current knowledge
# 2 and model performance

Cynthia H. Whaley[1], Kathy S. Law[2], Jens Liengaard Hjorth[3], Henrik Skov[3], Stephen R.
Arnold[4], Joakim Langner[5], Jakob Boyd Pernov[3,†], Rong-You Chien[6], Jesper H. Christensen[3],
Makoto Deushi[11], Xinyi Dong[6], Gregory Faluvegi[7,8], Mark Flanner[9], Joshua S. Fu[6], Michael
Gauss[10], Ulas Im[3], Louis Marelle[2], Tatsuo Onishi[2], Naga Oshima[11], David A. Plummer[1],
Luca Pozzoli[12,*], Jean-Christophe Raut[2], Ragnhild Skeie[13], Manu A. Thomas[5], Kostas
Tsigaridis[8], Svetlana Tsyro[10], Steven T. Turnock[14,4], Knut von Salzen[1], David W. Tarasick[15]
[1]Climate Research Division, Environment and Climate Change Canada, Victoria, BC, Canada
[2]LATMOS/IPSL, Sorbonne Université, UVSQ, CNRS,Paris, France.
[3]Department of Environmental Science/Interdisciplinary Centre for Climate Change, Aarhus University,
Frederiksborgvej 400, Roskilde, Denmark.
[4]Institute for Climate and Atmospheric Science, School of Earth and Environment, University of Leeds, Leeds,
United Kingdom.
[5]Swedish Meteorological and Hydrological Institute, Norrköping, Sweden.
[6]University of Tennessee, Knoxville, Tennessee, United States.
[7]NASA Goddard Institute for Space Studies, New York, NY, USA.
[8]Center for Climate Systems Research, Columbia University; New York, USA.
[9]Department of Climate and Space Sciences and Engineering, University of Michigan, Ann Arbor, MI, USA.
[10]Norwegian Meteorological Institute, Oslo, Norway.
[11]Meteorological Research Institute, Japan Meteorological Agency, Tsukuba, Japan.
[12]European Commission, Joint Research Centre, Ispra, Italy.
[13]CICERO Center for International Climate and Environmental Research, Oslo, Norway.
[14]Met Office Hadley Centre, Exeter, UK.
[15]Air Quality Research Division, Environment and Climate Change Canada, Toronto, ON, Canada.
†Now at: Extreme Environments Research Laboratory, École Polytechnique fédérale de Lausanne, 1951 Sion,
Switzerland
*now at, FINCONS SPA, Via Torri Bianche 10, 20871 Vimercate, Italy
*Correspondence to*: Cynthia H. Whaley (cynthia.whaley@ec.gc.ca)
**Abstract.** As the third most important greenhouse gas (GHG) after $CO_2$ and methane, tropospheric ozone ($O_3$) is
also an air pollutant causing damage to human health and ecosystems. This study brings together recent research
on observations and modeling of tropospheric $O_3$ in the Arctic, a rapidly warming and sensitive environment. At
different locations in the Arctic, the observed surface $O_3$ seasonal cycles are quite different. Coastal Arctic
locations, for example, have a minimum in the springtime due to $O_3$ depletion events resulting from surface
bromine chemistry. In contrast, other Arctic locations have a maximum in the spring. The 12 state-of-the-art
models used in this study lack the surface halogen chemistry needed to simulate coastal Arctic surface $O_3$ depletion
in the springtime, however, the multi-model median (MMM) has accurate seasonal cycles at non-coastal Arctic
locations. There is a large amount of variability among models, which has been reported previously, and we show
that there continues to be no convergence among models, nor improved accuracy in simulating tropospheric $O_3$
and its precursor species. The MMM underestimates Arctic surface $O_3$ by 5% to 15% depending on the location.
The vertical distribution of tropospheric $O_3$ is studied from recent ozonesonde measurements and the models. The
models are highly variable, simulating free-tropospheric $O_3$ within a range of +/- 50% depending on the model
and the altitude. The MMM performs best, within +/- 8% at most locations and seasons. However, nearly all
models overestimate $O_3$ near the tropopause (~300 hPa or ~8 km), likely due to ongoing issues with
underestimating the altitude of the tropopause and excessive downward transport of stratospheric $O_3$ at high
latitudes. For example, the MMM is biased high by about 20% at Eureka. Observed and simulated $O_3$ precursors



(CO, $NO_x$ and reservoir PAN) are evaluated throughout the troposphere. Models underestimate wintertime CO
everywhere, likely due to a combination of underestimating CO emissions and possibly overestimating OH.
Throughout the vertical profile (compared to aircraft measurements), the MMM underestimates both CO and $NO_x$
but overestimates PAN. Perhaps as a result of competing deficiencies, the MMM $O_3$ matches the observed $O_3$
reasonably well. Our findings suggest that despite model updates over the last decade, model results are as highly
variable as ever, and have not increased in accuracy for representing Arctic tropospheric.

## 1 Introduction

Tropospheric ozone ($O_3$) is the third most important greenhouse gas (GHG) after $CO_2$ and methane (IPCC, 2021),
and is an air pollutant causing damage to human health (WHO, 2013). It also causes damage to vegetation
following dry deposition to the surface (U.S. EPA, 2013). However, our knowledge about the sources and sinks
of tropospheric $O_3$ is still uncertain (AMAP, 2015; 2022; Gaudel et al., 2018), in particular in regions where fewer
observations exist, and where our understanding of key processes is still evolving. The Arctic is one such region
where few long-term measurements of $O_3$ exist and measurements of compounds that are important for producing
and destroying $O_3$ in the atmosphere are scarce at the surface and even more so in the free troposphere. Progress
has been made recently in terms of our understanding of certain processes and a picture is emerging about the
distribution of Arctic tropospheric $O_3$ as well as seasonal cycles and trends at different locations (e.g., Young et
al, 2018; Tarasick et al, 2019b). In particular, the connection between surface $O_3$ depletion episodes and halogens
is now well established (e.g., Simpson et al., 2007; Abbatt et al., 2012).
However, the role of natural cycles in the Arctic $O_3$ budget relative to $O_3$ produced from anthropogenic emissions
and how that relationship is changing in response to rapid warming in the Arctic are still uncertain. Arctic warming
and associated development in the Arctic are also driving changes in local anthropogenic emissions which could
already be leading to changes in the relative contributions of $O_3$ produced due to long-range transport of mid-
latitude anthropogenic emissions and $O_3$ produced from within or near-Arctic anthropogenic emissions. Increases
in emissions, such as from shipping (Gong et al., 2018) or boreal fires can affect Arctic air quality (Schmale et al.,

72     2018).

Ozone radiative forcing resulting from changes in tropospheric $O_3$ in the Arctic is highly sensitive to altitude. The
sensitivity of the Arctic $O_3$ vertical profile, and resultant forcing, from particular anthropogenic emission sources
and regions, vary substantially with altitude (Rap et al., 2015). Arctic surface $O_3$ may be most sensitive to European
or local sources (Sand et al., 2015; AMAP 2015; 2022), whereas emissions from North American and Asian
sources are more important in the mid- and upper troposphere (Monks et al., 2015; Wespes et al., 2012). Therefore,
a combination of varied source sensitivities in the vertical profile and the increased efficacy of longwave $O_3$ forcing
with altitude in the troposphere leads to a complex picture in terms of drivers of climate forcing by Arctic $O_3$. The
presence of temperature inversions in the Arctic lower troposphere may result in negative local forcing (Rap et al.,
2015; Flanner et al., 2018), in particular for local sources such as shipping (Marelle et al., 2018). Hence, to improve
the quantification of $O_3$ radiative effects in the Arctic there is a need first to assess model performance in terms of
seasonal cycles and vertical distributions. The annual mean vertical distributions of $O_3$ and CO were examined in
AMAP (2022) and Whaley et al. (2022) as compared to TES and MOPITT satellite retrievals. Those studies





showed good agreement between models and satellite measurements for $O_3$ in the free troposphere, where it is a
strong GHG.
This paper assesses the current state of knowledge about the dynamics of Arctic tropospheric $O_3$ and the ability of
a suite of current chemistry-transport and chemistry-climate models to simulate seasonal cycles of $O_3$ and selected
precursors. We first review our current understanding of sources and sinks of Arctic tropospheric $O_3$ in Section 2.
We summarize the models used in this study in Section 3 and the recent findings from satellite observations in
Section 4. We then examine the extent to which our understanding of Arctic tropospheric $O_3$ can explain observed
seasonal cycles at different surface sites in the Arctic and assess the ability of models to simulate observed
distributions (Section 5). We also examine vertical distributions of $O_3$ and its precursors and the extent to which
models are able to capture observed seasonal variations (Section 6). Finally, conclusions are presented in Section
7. Trends in Arctic tropospheric $O_3$ over the last 20-30 years and possible changes in seasonal cycles are presented
in a companion paper and compared to results from a subset of these models (Law et al., 2022).
**2. Arctic $O_3$: sources and sinks**
This section reviews tropospheric $O_3$ sources and sinks that are particularly relevant to the Arctic region, and many
of these processes are shown in the schematic in Figure 1.
**2.1 Ozone sources**
Tropospheric $O_3$ is a secondary air pollutant, which is not directly emitted but produced from the photochemical
reactions of anthropogenic and natural precursor emissions of VOCs, CO and $CH_4$ in the presence of $NO_x$. Besides
significant anthropogenic sources of these $O_3$ precursors, there are also important natural sources for these species,
such as boreal fires, lightning, vegetation and transport of $O_3$ from the stratosphere, which show marked seasonal
and inter-annual variations (Figure 1). Away from the surface, and in remote environments the tropospheric $O_3$
lifetime is around 20 days or more (Young et al., 2013), which facilitates the long-range transport of $O_3$ in the
troposphere. Production of $O_3$ from lower latitude emission sources and its subsequent transport to the Arctic is a
substantial source of Arctic tropospheric $O_3$, where the dry Arctic conditions and stably stratified atmosphere
further prolong the $O_3$ lifetime. In addition, the stratosphere-troposphere exchange of $O_3$ makes a substantial
contribution to the Arctic $O_3$ budget, where a lower tropopause height compared to the tropics facilitates the import
of stratospheric air masses rich in $O_3$. The weak in-situ $O_3$ formation in the Arctic relative to lower latitudes
increases the relative importance of this exchange. More recently, a new understanding has emerged regarding
contributions to Arctic surface $O_3$ from both anthropogenic and natural near-Arctic sources of $O_3$ precursors.
Downward transport of $O_3$ from the stratosphere is an important source of $O_3$ in the Arctic troposphere and may
be key in driving seasonality in Arctic tropospheric $O_3$ (Shapiro et al., 1987, Hess and Zbinden, 2013, Ancellet et
al., 2016). The Liang et al. (2009) modeling study suggests that in spring (March and April), most of the $O_3$ in the
Arctic upper troposphere originates from the stratospheric injection (78%) and that 20-25% of surface $O_3$
originates from direct injection of $O_3$ or the injection of $NO_y$ and secondary $O_3$ formation. Analysis of observations
by Tarasick et al. (2019a) is consistent with this picture. Global model simulations conducted as part of the Coupled
Model Intercomparison Project Phase 6, suggest an increase in near-surface $O_3$ over the Arctic during the 21st



century, driven by increased stratospheric $O_3$ import into the troposphere, particularly in winter (Zanis et al., 2022).
In contrast, during summer, the dominant source of Arctic tropospheric $O_3$ is in-situ production in the Arctic,
which in July contributes more than 50% of $O_3$ in the Arctic boundary layer and 30-40% in the free troposphere
(Walker et al., 2012). This study also showed that the transport of peroxyacetyl nitrate (PAN) from lower latitudes
is the dominant source of $NO_x$, driving in-situ $O_3$ production at the surface in late spring and early summer.
Methane ($CH_4$) is a key precursor for tropospheric $O_3$, via its oxidation in the presence of sufficient $NO_x$.
Anthropogenic $CH_4$ emissions are estimated to be responsible for around half of the global radiative forcing due
to tropospheric $O_3$ from pre-industrial to present-day (Stevenson et al. 2013). Fiore et al. (2008) estimated that
anthropogenic $CH_4$ emissions contribute 15% to the annual average total global $O_3$ burden (including natural and
anthropogenic sources). Based on parameterised source-receptor sensitivities for a range of CMIP6 SSP scenarios,
Turnock et al. (2019) illustrated the significant contribution of $CH_4$ to future $O_3$ concentration reductions at high
latitudes. Using a similar approach, based on parameterised responses to $O_3$ precursor emission perturbations found
that $CH_4$ dominates the sensitivity of Arctic $O_3$ to anthropogenic emissions (AMAP, 2015). $CH_4$ accounts for
approximately 40% of the Arctic $O_3$ response to precursor emission perturbations (AMAP, 2015).
Import of $O_3$ and its precursors from lower latitudes associated with episodes of long-range transport of
anthropogenic or biomass burning pollution leads to enhancements in Arctic tropospheric $O_3$ (Wespes et al., 2012;
Monks et al., 2015; Ancellet et al., 2016). Whilst very low levels of $NO_x$ within the Arctic, away from local
sources, often limit local $O_3$ production, the release of $NO_x$ from thermal decomposition of peroxy-acetyl nitrate
(PAN) (an important $NO_x$ reservoir) imported from lower latitudes, can lead to in-situ production of $O_3$,
particularly in the warmer Arctic summer lower troposphere (Wespes et al., 2012; Arnold et al., 2015).
Investigation of long-range transport of $O_3$ precursors has shown efficient export of PAN from East Asia to the
North Pacific, with relative contributions to long-range $O_3$ transport of 35% in spring and 25% in summer (Jiang
et al., 2016). Ship observations over the Arctic and Bering Seas also identified events of long-range pollution
transport with enhancements in $O_3$ (Kanaya et al., 2019).
Recently, there has been progress in improving knowledge of local $O_3$ precursor sources. Surface $O_3$ in summer is
already influenced by shipping $NO_x$ emissions along the northern Norwegian coast (Marelle et al., 2016; Marelle
et al., 2018) and the Northwest Passage (Aliabadi et al., 2015). Marelle et al., (2018) showed for a 2050 scenario,
including diversion shipping in the Arctic, shipping would become the main surface $O_3$ source. Tuccella et al.
(2017) showed that background $O_3$ is influenced by emissions downwind of oil and gas extraction platforms in the
southern Norwegian Sea. Granier et al. (2006) predicted the increase in shipping activity during the summer would
increase $O_3$ levels by a factor of 2-3 in the coming decades, with maximum increases (> 20 ppbv) occurring in the
Canadian Archipelago, Beaufort Sea, central Arctic Ocean, and the Siberian sector of the Arctic.
Natural sources of Arctic tropospheric $O_3$ precursors include lightning $NO_x$, emissions of $NO_x$ and reactive VOCs
from the snowpack (Honrath et al., 1999; Guimbaud et al., 2002; Hornbrook et al., 2016; Pernov et al., 2021), and
natural emissions of VOCs from high latitude vegetation (Holst et al., 2010; Ghirardo et al., 2020), and the Arctic
Ocean (Mungall et al., 2017). Evidence from both observations and models suggests that boreal fires are also an
important source of $O_3$ precursors and $NO_x$ reservoir species like PAN, in spring and summer, with impacts on
Arctic $O_3$ (Thomas et al., 2013; Arnold et al., 2015; Viatte et al., 2015; Ancellet et al., 2016).



**2.2 Ozone sinks**
Photochemical loss of $O_3$ is mainly via photolysis in the presence of water vapor or direct reaction of $O_3$ with
hydroperoxyl or hydroxyl radicals ($HO_2$ or $OH$). Photochemical destruction involving the hydroperoxyl radical
($HO_2$) may be particularly important in the Arctic where water vapor abundances are low (Arnold et al. 2015).
Where local emission sources give rise to high $NO_x$ concentrations in urban regions or regions of shipping activity,
$O_3$ loss via titration with NO can be dominant (Thorp et al., 2021; Raut et al., 2022). Dry deposition of $O_3$ and its
precursors to ice and ocean surfaces is slower than to vegetated terrestrial surfaces (Figure 1). Van Dam et al.
(2016) reported $O_3$ dry deposition velocities that were 5 times higher over Arctic snow-free tundra in the summer
months at Toolik Lake (northern. Alaska) compared to the snow-covered ground. Dry deposition, combined with
possible chemical loss (e.g. involving Biogenic-Volatile Organic Compounds, BVOCs) producing lower $O_3$
concentrations during stable (lower light) night conditions may explain the different diurnal cycle observed at this
tundra site compared to Arctic coastal locations. Interestingly, gradient studies at Barrow showed a positive
gradient with height during $O_3$ depletion events (ODE) and atmospheric mercury depletion events (AMDE)
suggesting that $O_3$ was removed at the surface due to fast photochemical reactions at or close to snow surfaces
initiated by the release of halogen species (Skov et al., 2006). During ODEs at Arctic sites in the Canadian
archipelago (Alert, Resolute, and Eureka), vertical profiles show ozone is typically uniformly depleted in the
boundary layer whereas a positive gradient is observed above the boundary layer (Tarasick and Bottenheim, et al.,

176      2002).

During Arctic spring, photochemical cycling of halogens in so-called 'bromine explosion' events leads to rapid
depletion of surface $O_3$ to low or near-zero concentrations (Barrie et al. 1988; Simpson et al. 2007). These
phenomena are most commonly observed at Arctic coastal locations in March/April and attributed to bromine
(halogen) sources linked to Arctic sea ice, coupled with stable surface temperature inversions (e.g. Figure 1;
Hermann et al., 2019). Interestingly, Yang et al. (2020) and Huang et al. (2020) were able to explain major
depletion events in a model study by introducing the wind-induced release of bromine from the snowpack.
However, the models could not explain the depletion events observed at low wind speeds. Swanson et al. (2022)
used the GEOS-Chem model to show that both blowing snow and the snowpack are important sources of bromine
during the spring. Figure 2 shows the vertical extent of low $O_3$ episodes observed by lidar at Eureka in northern
Canada. On May 7th, low $O_3$ concentrations were observed and back trajectories showed that air masses came in
from the ice-covered Arctic Ocean and had been in contact with the surface multiple times during the previous 6
days, whereas the concentrations were high on May 9, when air came down from the mountains located to the
south (Seabrook and Whiteway, 2016). Peterson et al. (2018) showed that active halogen chemistry and related $O_3$
depletion can also occur up to 200 km inland over snow-covered tundra in Alaska. Simpson et al. (2018) reported
high levels of bromine oxide (BrO) at Utqiaġvik (previously known as Barrow, Alaska) occurring earlier in
February in air masses originating from the Arctic Ocean polar night. Their findings suggest a dark wintertime
source of reactive bromine (halogens) that could feed halogen photochemistry at lower latitudes as the sun returns.
In addition, whilst earlier studies proposed indirect evidence that $O_3$ and gaseous elemental mercury ($Hg^0$) is
removed by reaction with Br atoms (e.g. Skov et al. 2004; Skov et al. 2020; Dastoor et al. 2008), Wang et al.
(2019) showed, for the first time, a direct connection between $O_3$ and $Hg^0$ with atomic bromine (Br) during $O_3$ and
$Hg^0$ depletion episodes at Utqiagvik, on the north coast of Alaska (see Figure 3) where $O_3$ and $Hg^0$ are removed



in competing reactions with Br. Here, the Br/BrO ratio anti-correlates with $O_3$ concentrations and box modeling
confirms that $O_3$ is removed by Br.
This result is significant since the main source of halogens in the Arctic is the release from refreezing snow and
ice, snow blowing over sea ice, heterogeneous reactions of aerosol particles, and snowpack recycling (Petersen et
al. 2016; Peterson et al., 2017, Wang et al., 2017; Yang et al. 2020). Burd et al. (2017) found a strong relationship
between the end of the reactive bromine season and snowmelt timing. In the future, continued decreases in Arctic
sea ice extent or the relative distributions of multi-year/seasonal sea-ice cover, coupled with increases in the length
of the snow-free season over land could influence the magnitude and seasonality of $O_3$ sinks via changes in halogen
fluxes or dry deposition fluxes to tundra/ocean rather than snow/ice surfaces.
**3. AMAP models and simulations**
To evaluate our process understanding of controls on the Arctic tropospheric $O_3$ budget and distribution, we
evaluate a suite of model simulations. Twelve atmospheric models participated in this study; 7 chemical-transport
models (DEHM, EMEP-MSC W, GEOS-Chem, MATCH, MATCH-SALSA, OsloCTM, WRF-Chem) and 5
chemistry-climate models (CESM, CMAM, GISS-E2.1, MRI-ESM2, and UKESM1), with simulations of the years
2014-2015 for comparisons to observations. All models used the same set of anthropogenic emissions called
ECLIPSEv6b (AMAP 2022), though had different sources for fire, biogenic emissions, and meteorology (see
Table S1). All participating models prescribe $CH_4$ concentrations based on box model results, which are, in turn,
based on the ECLIPSEv6b anthropogenic $CH_4$ emissions, and various assumptions on natural $CH_4$ emissions
(Olivié et al., 2021; Prather et al., 2012). Models then allow $CH_4$ to take part in photochemical processes. The
participating models have varying degrees of spatial resolution and chemical complexity; air quality-focused
models, such as DEHM, EMEP MSC-W, GEOS-Chem, MATCH, and WRF-Chem have detailed $HO_x$-$NO_x$-
hydrocarbon $O_3$ chemistry, with speciated volatile organic compounds (VOCs), and secondary aerosol formation,
and tend to run at higher resolution. The earth system models GISS-E2.1, MRI-ESM2, and UKESM1 also contain
this level of tropospheric chemistry, though run globally at coarser resolution. Whereas, climate-focused models
like CMAM, run at a coarse resolution and have simplified tropospheric chemistry in order to be able to run for
long periods. For example, CMAM's tropospheric chemistry consists only of $CH_4$-$NO_x$-$O_3$ chemistry, with no
VOCs.
As mentioned above, Arctic tropospheric $O_3$ is heavily influenced by imports from the stratosphere. The models
vary, too, in their representation of the stratosphere. Only a subset of participating models have a fully simulated
stratosphere. CMAM, MRI-ESM2, GISS-E2.1, OsloCTM, and UKESM1 contain relatively complete stratospheric
$O_3$ chemistry ($NO_x$, $NO_x$, $Cl_x$, $Br_x$ chemistry that controls stratospheric $O_3$). Other models have a simplified
stratosphere, such as GEOS-Chem which has a linearized stratospheric chemistry scheme (LINOZ, McLinden et
al., 2000), and WRF-Chem which specifies stratospheric concentrations from climatologies. Finally, several
models have no stratosphere or stratospheric chemistry at all (e.g., DEHM, and EMEP MSC-W). Most atmospheric
models, including all of the models in this study, do not yet contain Arctic tropospheric bromine chemistry, and
thus cannot simulate the surface-level bromine-driven $O_3$ depletion events that occur during spring. However,


there are research versions of some models which are starting to contain this chemistry (e.g., Parrella et al., 2012;
Falk and Sinnhuber, 2018; Badia et al., 2021)
These same 12 model simulations were also evaluated against a different set of measurements in AMAP (2022)
and Whaley et al. (2022). Those studies focused on many SLCF species over the Northern Hemisphere and
generally reported model biases for the annual mean concentrations. They found that all models overestimated
surface $O_3$ concentrations at mid-latitudes, but that there were both over- and underestimation in the Arctic.
Particularly, models overestimated surface $O_3$ in the western Arctic (e.g., Alaska), particularly in the summertime,
but were better able to simulate the surface $O_3$ seasonal cycle in the eastern Arctic (e.g., northern Europe). They
also found that model biases were small throughout the free-troposphere when compared to remote measurements
from the Tropospheric Emission Spectrometer (TES) satellite instrument.
In the next sections, these models are compared with observations of $O_3$ and its precursors either individually, or
as the multi-model median (MMM) - whereby the median of all 12 atmospheric models is shown unless otherwise
noted. The model output was selected from the model grid box that contains the latitude and longitude of the
observation location without any spatial interpolation.
**4. Arctic-wide tropospheric distributions from satellite data**
Despite the potential limitations of some satellite data products at high latitudes, several studies have exploited
satellite observations to investigate tropospheric $O_3$ and precursor distributions and trends relevant to the Arctic.
Pommier et al. (2012) presented IASI retrievals of 0-8 km and 0-12 km sub-column $O_3$ for the Arctic in spring and
summer 2008. These showed widespread enhancements in spring-time (Mar-Apr) tropospheric $O_3$ column
compared with summer (Jun-Jul), particularly over northeast Siberia, northern Canada and the Arctic Ocean.
Generally, good agreement with in-situ aircraft profiles was demonstrated, but negative IASI biases were found
compared with aircraft data in the lower troposphere, due to low thermal contrast in the Arctic boundary layer.
Wespes et al., (2012) showed that IASI was able to detect enhancements in mid-latitude sourced $O_3$ enhancements
during summer at the edge of the Arctic, but also showed a lack of sensitivity over snow and ice surfaces,
potentially resulting in missing some $O_3$ enhancements. Sodemann et al. (2011) analyzed the cross-polar transport
of a large pollution plume originating from Asia during the summer of 2008 using IASI CO retrievals. IASI was
able to detect features and structures of the plume consistent with in-situ aircraft data.
Satellite observations are also useful in evaluating the sources and export of $O_3$ precursors from mid-latitude source
regions and their subsequent transport to the Arctic. Tropospheric NO2 columns measured from OMI have been
used to detect enhancements and trends in $NO_x$ emissions due to gas flaring in high latitude (up to 67°N) areas of
Russia and North America (Li et al., 2016). Assessment of a suite of chemical transport models using OMI
tropospheric $NO_2$ columns for summer 2008 showed a potential overestimate in $NO_2$ over biomass burning regions
in eastern Siberia, with lower biases over European and North American source regions, and model under-
estimates over China (Emmons et al., 2015). A comparison of regional model-simulated tropospheric $NO_2$ columns
with observations from the OMI satellite instrument suggests potential underestimates in anthropogenic $NO_2$
emissions over high latitude Siberia and the Russian Arctic (Thorp et al., 2021). Monks et al., (2015) exploited
limited profile information from MOPITT CO retrievals to evaluate relationships between CO seasonal cycles in



the lower and upper troposphere over the Arctic and mid-latitude source regions. Atmospheric Infrared Sounder
(AIRS) CO retrievals from 2007 to 2018 have been used to characterize atmospheric circulation patterns coincident
with pollution enhancements during Arctic spring (Thomas et al., 2021), and IASI CO column measurements have
been used to analyze transport pathways for Asian anthropogenic pollution to the Arctic (Ikeda et al., 2021). Osman
et al. (2016) constructed three-dimensional (5° x 5° 1 km) gridded climatologies of CO via a domain-filling
trajectory mapping technique based on MOZAIC-IAGOS in situ measurements of commercial aircraft flights.
These climatologies agreed well using forward and backward trajectories (< 10 % difference for most cases) and
against vertical measurements from MOZAIC-IAGOS not included in the climatologies. These climatologies were
compared with CO retrievals from MOPITT, small biases were found in the lower troposphere while differences
of ~20 % were found between 500 and 300 hPa, which declined throughout the study (2001-2012). Inter-annual
variability in PAN retrieved by TES over Eastern Siberia for April 2006-2008 was documented by Zhu et al.,
(2015), and it was shown to be largely controlled by boreal fire emissions at this time of year. More recently, PAN
data from the TES instrument was used to help characterize Asian influence on exported PAN and downwind $O_3$
production (Jiang et al., 2016). A temperature-dependent high bias in TES PAN was found at cold temperatures
over high latitudes.
In both Chapter 7 of the 2022 AMAP SLCF report (AMAP, 2022) and Whaley et al. (2022), data from satellite
instruments, TES, ACE-FTS, and MOPITT are used to evaluate modeled $O_3$, $CH_4$, and CO in the Northern
Hemisphere. They showed that model biases for $CH_4$ were small, though tended to be negative in the Arctic due
to a lack of north-south gradient in the prescribed global distribution. Model biases were also negative for free-
tropospheric $O_3$, however, it was by approximately the same amount that TES $O_3$ retrievals have been shown to
be biased high by Verstraeten et al. (2013). The ACE-FTS comparison for $O_3$ showed good agreement but had
higher model biases around 300-100 hPa in Whaley et al. (2022) and AMAP (2022). The MOPITT CO
comparisons in AMAP (2022) showed that all models' CO are biased low over land in the mid-latitudes, but biased
high over the oceans at lower latitudes. Monks et al. (2015) discussed that models had high biases in the outflow
from Asia, and low biases north of there due to lack of transport. The Quennehen et al. (2016) study also suggested
that summertime CO transport out of Asia is too zonal. This could explain some of the underestimations in the
Arctic CO in the mid-troposphere.
**5. Arctic surface $O_3$ and precursors: seasonal cycles**
In the high Arctic, there is very little diurnal variation in surface $O_3$, most likely because the local and regional
photochemistry is of limited importance most of the time and due to the 24-hour daylight during Arctic spring,
summer and Autumn as well as the polar night during winter, see earlier. For High Arctic sites, the seasonal
dynamics of $O_3$ can be explained mostly by long-range transport and particularly in the winter and springtime,
intrusion from aloft, see Figure 5a. Moving southwards to the Polar Circle a clearer diurnal pattern is evident
caused both by the seasonal behavior of vertical mixing, deposition, transport, and local chemistry (references) as
the stations on the Scandinavian peninsula,  and Denali central Alaska.



### 5.1 Surface Ozone

Seasonal differences in the Arctic are important because of differences between the local meteorological conditions, as well as atmospheric transport, in the warm and the cold seasons and seasonal variations in $O_3$ sources and sinks. Surface $O_3$ at remote midlatitude sites with limited influence from local and regional anthropogenic $O_3$ precursor emissions have been found to frequently exhibit a characteristic seasonal cycle with peak values during spring and a minimum in the summer, while sites with high exposure to $O_3$ from anthropogenic precursors have summer time $O_3$ maxima (Monks 2000; Parrish et al. 2013, 2019; Gaudel et al., 2018). The spring maxima has been explained by stratospheric intrusions as well as enhanced photochemical formation during this period of the year. The summer minima, e. g. observed at the Mace Head site (Derwent et al., 1998, 2013, 2020), which is strongly influenced by marine air, appears to be explained by photochemical destruction in the absence of anthropogenic precursors. Seasonal cycles at Arctic stations are not extensively discussed in the literature, but it is evident that the halogen chemistry discussed above, most frequently observed at high Arctic coastal stations, leads to a significant reduction during the springtime (e.g. Oltman and Komhyr, 1986; Tarasick et al., 1995; Monks et al., 2015). Regarding the more southerly Arctic and near-Arctic sites, a latitudinal gradient has been observed in the timing of the spring $O_3$ maximum. Anderson et al. (2017) found that monthly mean observed near-surface $O_3$ concentrations at background sites in Sweden from 1990 to 2013 had a maximum in spring, but the most northerly stations experienced their maximum in April while the more southerly ones in May.

In order to get an overview of the annual $O_3$ cycles at different types of Arctic surface measurement sites, we have calculated the monthly medians and interquartile range for the period 2003-2019 for a series of sites. A map of the stations as well as their coordinates and elevation can be seen in Figure 4. Figure 5 illustrates the range of seasonal cycle behavior observed in the Arctic at different measurement sites and shows different seasonal cycles depending on location.

### 5.1.1 High Arctic sites

Figure 5a shows that the seasonalities in $O_3$ at Villum, Barrow/Utqiagvik, Alert, Tiksi and Eureka are similar: They have a local minimum in spring due to the occurrence of ODEs, a slight increase/recovery in June and a second minimum in July due to surface removal and photochemical degradation of $O_3$. These stations are located at high latitude coastal sites close to sea level. During winter, $O_3$ reaches a maximum, due to an absence of photochemical degradation of $O_3$, vertical mixing is suppressed during polar night since the Arctic boundary layer is often highly stratified, thus hampering removal by dry deposition (Esau and Sorokina, 2016).

### 5.1.2 Near Arctic Circle sites

The characteristic seasonal variations of surface $O_3$ measured at stations close to the Arctic Circle are shown in Figure 5b. The stations are Karasjok and Tustervatnet in Norway, Esrange in Sweden, Pallas in Finland and Denali in Alaska. The sites in Figure 5b, which are not influenced by ODEs, exhibit a yearly cycle that is more similar to lower latitude European stations at remote locations. Here, surface $O_3$ exhibits a late spring maximum which is attributed to photochemical production and transport of $O_3$ from the stratosphere (Monks, 2000). The largest differences between the stations are mainly found during the summer months, most likely due to the influence of local sources on photochemical $O_3$ production.





Kårvatn in Norway has an unusual behavior with an O₃ maximum in March, possibly due to the local conditions:
The site often shows a pronounced diurnal cycle in O₃ due to the location at the bottom of a valley that causes
strong inversions leading to an enhanced impact of dry deposition at night on surface O₃ (Aas et al., 2017).
Hurdal in Norway is included as an example of a more southerly Scandinavian non-Arctic station, which has an
annual variation with a minimum in October while the more northerly stations have minima between July and
September (Figure 5c), this difference may be explained by a stronger influence of local air pollution at Hurdal.
At Hurdal, winter O₃ concentrations are particularly low, probably also in this case due to the influence of local
emissions which in this period leads to the removal of O₃ by the reaction with NO.

### 352    5.1.3 Other sites: Inland high-elevation

Summit (located in the free troposphere on the Greenland Ice Sheet) is much less affected by bromine chemistry
originating from sea ice or other low altitude processes than the coastal High Arctic sites (Huang et al. 2017).
Consequently, the seasonal variation is different with a clear maximum in May, a minimum in September, the
higher concentrations compared to other surface stations can be explained by the high sensitivity to stratospheric
O₃ enriched air (Monks et al., 2015) at this high elevation (3211 masl) site. Short episodes of depletion have been
reported (Brooks et al. 2011) but they do not appear to affect the monthly mean values substantially as shown in
Figure 5c.
Zeppelin, although a high Arctic site, is located on a mountain ridge at 474 masl and thus experiences free
tropospheric air masses more often compared to sea level sites. For this reason, it is less influenced by ODEs and
consequently does not have an O₃ minimum in spring like the other high Arctic coastal stations (Figure 5c). That
said, the occasional ODE has been reported there by Lehrer et al. (1997) and Ianniello et al. (2021).
We also note that surface O₃ can be influenced by local anthropogenic emissions such as shipping (e.g. Marelle et
al., 2016, Aliabadi et al., 2015) or oil field emissions (McNamara et al., 2019). McNamara et al. (2019) discussed
potentially important interactions between local anthropogenic NO$_x$ emissions from the Barrow settlement or the
Prudhoe Bay oil extraction facilities in northern Alaska and snowpack (chlorine) chemistry leading to elevated
concentrations of nitrogen-containing compounds (e.g. N₂O₅, HO₂NO₂), with implications for Arctic tropospheric
O₃. Therefore, while none of the Arctic sites currently exhibit summertime surface maxima due to photochemical
production, as often observed in polluted locations further south, this may change in the future with increasing
local anthropogenic emissions (e.g. Marelle et al.2018).
He et al. (2016) measured O₃ and black carbon on a ship cruise to the Arctic Ocean (31.1°N to 87.7°N and 9.3°E–
90°E to 168.4°W) from June to September 2012. Comparing the observed O₃ concentrations to those measured at
Barrow showed no statistically significant differences, the authors suggest that coastal stations between July and
September may be representative of the entire Arctic but this hypothesis requires further investigation. Indeed, our
results show significant differences in the O₃ seasonal cycles at different Arctic locations depending on whether
they were coastal, in-land, or high elevation.


### 379   5.2 Surface $O_3$ model evaluation

It has been found that halogen chemistry, stable boundary layers, and dry deposition explained differences between
measured and modeled $O_3$ concentrations, as demonstrated by Kanaya et al. (2019) who performed measurements
of CO and $O_3$ during several ship cruises in the Bering Sea and the Arctic Ocean in September (2012 to 2017).
None of the models in our study contain surface halogen chemistry but they also display highly variable agreement
in their surface $O_3$ seasonal cycles. Figure 6 shows the seasonal cycle from the models and observations averaged
for 2014-15 at several Arctic observation locations. Since the models do not contain surface-level bromine
chemistry, at locations like Alert and Barrow/Utqiagvik, they do not capture the springtime minimum in $O_3$. Some
models (e.g. UKESM1) greatly underestimate wintertime $O_3$. This may be related to deficiencies in boundary layer
mixing or an overly shallow boundary layer depth, resulting in the overly active titration of $O_3$ by NO near $NO_x$
emission sources and subsequent underestimation of Arctic surface $O_3$. However, other model deficiencies could
also play a role, including dry deposition and $NO_x$ lifetime. Indeed, Barten et al. (2021) found that overestimation
of oceanic $O_3$ deposition can explain some differences between modeled and measured surface $O_3$ in the High
Arctic. Some models in Figure 6 do not agree on the timing of the springtime peak, with CMAM, DEHM, and
GISS-E2.1 peaking in April, and EMEP-MSC-W and MRI-ESM2 peaking in May/June. The same groupings of
models display different $O_3$ behavior at the end of the year (October-December), with CMAM, DEHM, and GISS-
E2.1 all correctly simulating an increase in $O_3$, and EMEP-MSC-W and MRI-ESM2 having a decrease. All models
agree better with observations and each other on summertime surface $O_3$ abundance at all locations, and on the
full seasonal cycle at Summit, the high-elevation background location. The large range of modeled surface $O_3$ is
similar to previous model studies (Shindell et al.,2008; Monks et al., 2015, Gaudel et al., 2018). Despite the large
range in model performance, the overall average negative $O_3$ bias, and the seasonality in model bias at
Barrow/Utqiagvik and Summit, are consistent with these previous studies. The comparisons highlight little change
in the skill of models in simulating Arctic surface $O_3$ over the past decade.
These particular model simulations have been evaluated in Whaley et al. (2022), where they grouped all western
Arctic (defined as lat > 60°N, and lon < 0°) and eastern Arctic (lat > 60°N, lon > 0°) $O_3$ measurements together, and
showed the range in modeled and measured seasonal cycles for those two regions. That analysis included
additional locations at lower latitudes, thus their results emphasized that some models overestimated summertime
$O_3$ in the western Arctic. Otherwise, the results from that study are consistent with what we report here.

### 407   5.3 Ozone precursors

$NO_x$ monitors have been used at several Arctic sites but in a study at Zeppelin, it was shown that most of the $NO_x$
was in the form of the $NO_x$ reservoir species PAN (Beine et al., 1997; Beine and Krognes, 2000). We evaluate and
discuss PAN in Section 6.3 from aircraft measurements. There are only limited sources for $NO_x$ in the Arctic and
the lifetime of $NO_x$ is in the order of a day. Whaley et al. (2022) evaluated surface $NO_x$ volume mixing ratios and
found that these models underestimated surface $NO_2$ by -59% at low-Arctic latitudes that were mostly around
60ºN.
The dominant source for $NO_x$ is long-range transport of dominantly PAN (Beine and Krognes; 2000), and
particulate bound $HNO_3$ followed by reactivation in the Arctic by thermal decomposition and photoreduction
processes, respectively. Kramer et al. (2015) determined at Summit from July 2008 to July 2010 that PAN
accounted for 295 ppt, and NO$_x$ for 88 ppt. In a more recent study, Huang et al. (2017) found in the period July
2008–June 2010, PAN and NO$_x$ were maximum in spring at about 250 ppt and 25 ppt, respectively, and in summer
75 ppt and 20 ppt, respectively. Beine and Krognes, (2000) measured PAN at Zeppelin Mountain between 1994
and 1996. They found median values were lowest in summer at 89.4 ppt and highest in spring at 222.6 ppt. HNO$_3$
in the gas phase is in general very low (Wespes et al., 2012). Particulate bound nitrate – potentially a significant
source of NO$_x$ in the atmosphere and snow pack – is close to the detection limit in summer and up to 124.7 ng N
m-3 in winter e.g. at Villum (Nguyen et 2013).
In general, NMVOC concentrations in the Arctic are low and thus their photo-oxidation has only a limited impact
on O$_3$. There is a series of studies dedicated to specific source regions and emission sources. There is one long
term measurement study by Gautrois et al. (2003); studies focusing on long-range transport (Stohl, 2006; Harrigan
et al., 2011), snowpack emissions (Boudries et al., 2002; Dibb and Arsenault, 2002; Guimbaud et al., 2002; Barret
et al., 2011; Gao et al., 2012) and shipborne measurements (Sjostedt et al., 2012 and Mungall et al., 2017). The
Gautrois et al. (2003) study reported long-term VOC concentrations for Alert, NU,  they found yearly levels of
ethane, propane and toluene are 1.7 ppbv, 0.6 ppbv and 26 pptv, respectively. For comparison, mixing ratios of
ethane, propane, and toluene in China ranged from 3.7-17 ppbv, 1.5-20.8 ppbv, 0.4-11.2 ppbv, respectively
(Barletta et al., 2005).
Pernov et al. (2021) measured organic O$_3$ precursors online with a PTR-ToF-MS at Villum from April to October
2018. Sources were apportioned with Positive Matrix Factorization During the late spring, the Arctic haze factor
was a source of oxygenated VOCs (OVOCs) arising from long-range transport of anthropogenic emissions whilst
during summer OVOCs and DMS originated from the Marine cryosphere factor. During autumn, the Biomass
burning factor peaked in importance and was dominated by acetonitrile. The most abundant compound during the
campaign was acetone with a mean mixing ratio of 0.6 ppbv, for benzene 0.027 ppbv and DMS 0.046 ppbv. In the
future, local NMVOC emissions might increase from both natural and anthropogenic sources due to the retreating
sea ice with more biological activity and more industrial activity and shipping affecting future levels of O$_3$.
Figure 7 shows the observed and simulated seasonal cycle of CO at Zeppelin and Utqiagvik/Barrow. Simulated
CO ranges about 50 ppbv across models, and all models underestimate surface CO at these sites. The low model
biases are dominated by the winter and spring months. The 2014-15 annual multi-model median (MMM) bias is -
11% and -16% at Zeppelin and Barrow/Utqiagvik, respectively. Figure 7 shows that for the first 6 months of the
year, the MMM is 20-30% too low, but that in the summer, the MMM is much closer to observations. These CO
results are very similar to those found in previous multi-model studies (Shindell et al., 2008; Monks et al., 2015;
Whaley et al., 2022). Similar to O$_3$, these results imply little change in the skill of models in simulating Arctic
surface CO over the past decade. The modeled CO underestimations are well-reported in the literature, and
attributed either to a lack of CO from combustion sources in the emission inventories (e.g., Kasibhatla et al., 2002;
Pétron et al., 2002; Jiang et al., 2015), or to errors in OH, which impact the lifetime of CO (e.g., Monks et al.,
2015; Quennehen et al., 2016). Indeed both may be at cause here, as the anthropogenic CO emissions from
ECLIPSEv6b are lower than those in the CMIP6 emission inventory, neither of which have taken into account the
reported discrepancies from top-down emissions studies (Kasibhatla et al., 2002; Pétron et al., 2002; Jiang et al.,



2015, Miyazaki et al., 2020). Monks et al. (2015) showed that models with lower global mean OH concentrations
produced smaller underestimates in Arctic surface CO and that models with larger underestimates in CO over mid-
latitude source regions also had larger underestimates in Arctic CO. Emmons et al. (2015) showed that the models
with larger tropospheric OH also had higher photolysis rates of $O_3$ to $O^{1D}$ and that there was also some relationship
between higher photolysis rates and lower cloud cover fraction in some models. Previous multi-model results have
also shown that variability in model water vapour abundance in the Arctic appeared to be the leading driver of
model variability in OH, despite being much less important at lower latitudes (Monks et al., 2015). Evaluating OH
and water vapour is unfortunately beyond the scope of our study.
Methane has more than doubled since preindustrial times (from 0.8 ppmv to 1.8 ppmv) and the photooxidation of
methane in the presence of $NO_x$ is a source of $O_3$. Thawing permafrost and release from organic deposits in shallow
Arctic Ocean waters in a warmer climate presents a new source of methane (Isaksen et al. 2014).  The models of
this study prescribed $CH_4$ concentrations, including their increasing trend, and they were found to have a small
bias of ~2% in Whaley et al. (2022) compared to surface and satellite measurements. Going forward, models are
starting to simulate CH4 explicitly from emissions, and this will be important for simulating future changes in
Arctic tropospheric chemistry.
**6. Vertical distributions of $O_3$ and precursors in the Arctic**
Observations and models have both demonstrated extensive layering of pollution signatures in the Arctic
troposphere vertical profile, associated with varying air mass origins with altitude (Zheng et al., 2021; Willis et
al., 2019). Large-scale isentropic transport pathways result in air masses from warmer more southerly latitudes
being imported into the Arctic upper troposphere, while emissions from cooler northerly latitudes enter the Arctic
near the surface and in the lower troposphere (Stohl, 2006). The presence of the Arctic dome during winter
essentially shuts off access to the Arctic surface to air mass import from southerly mid-latitudes, while it facilitates
efficient low-level transport of emissions from Northern Eurasia and Russia to the Arctic surface, giving rise to
the well-known Arctic haze (Shaw, 19995). In practice, this large-scale dynamical control on long-range transport
to the Arctic gives rise to a well-characterized vertical dependence of source region sensitivities for $O_3$ and
precursors through the Arctic troposphere, where emissions from South and East Asia have the most influence in
the Arctic upper troposphere, emissions from North America have the most influence in the Arctic mid-
troposphere, and northern Eurasian and Russian emissions dominate at the surface (in addition to local influences)
(Wespes et al., 2012; Monks et al., 2015). As mentioned in Section 1, this vertical layering and changes in the
efficacy of $O_3$ radiative forcing with altitude has implications for the sensitivity of Arctic tropospheric $O_3$ forcing
to regional emission perturbations.
Despite evidence for extensive vertical layering in the Arctic troposphere, and the potential for highly varying
source contributions with altitude, aside from a limited set of regular $O_3$ sonde profiles, there is a severe lack of
observations available on the vertical distribution of $O_3$, and particularly its precursors, in the Arctic troposphere.
There is an especially poor constraint on seasonal and interannual variability in $O_3$ precursor profiles. In this
section, we make use of available vertical profile measurements of $O_3$ and its precursors to document our
understanding of Arctic tropospheric $O_3$ profiles, and to evaluate model-simulated vertical profiles of $O_3$ and
precursors.





### 6.1 Ozonesondes

Ozone soundings provide a long-term record of Arctic $O_3$ through the depth of the troposphere. Since 1966, weekly soundings have been available at Resolute and since the 1980s regular soundings, typically once a week, have been available from 6 stations north of 60ºN (Figure 4, Table S.2). All of these stations are located in the sector 95ºW to 27ºE meaning that regular soundings are lacking in a large sector of the Arctic. The measurements are conducted using the balloon-borne Electrochemical Concentration Cell (ECC) ozonesondes, typically reaching an altitude of about 30 km. Random uncertainties in tropospheric measurements are about 5%, and biases reported from field and laboratory comparisons to UV reference photometers are 1.0±4.4% in the lower troposphere and 5.3±4.4% in the upper troposphere (Tarasick et al., 2019b). Mean observed concentrations have a minimum close to the surface and then gradually increase throughout the troposphere by about 50% and then increase sharply going into the upper troposphere and lower stratosphere (Figure 8 and Figure S.1-2). Observed seasonal cycles in the Arctic troposphere generally show a maximum in spring and summer and a minimum in fall and winter (Paper 2).

Christiansen et al. (2017) examined long-term ozonesonde records at 9 Arctic stations reporting consistent seasonal cycles as a function of altitude between sites with later maxima in the mid-troposphere compared to the surface layers and upper troposphere.

### 6.2 Model evaluation against ozonesondes

Figure 8 shows a comparison of the ozonesonde measurements at Eureka to the simulations from the 12 participating models for the annual and seasonal averages for the years 2014-15. In the supplement (Figure S.2), model-measurement comparisons at other Arctic locations are shown. Generally, the models are highly variable, ranging +/-50% of the measured $O_3$ profiles at most seasons and locations. However, the MMM performs well and is within +/-8% throughout most of the troposphere. However, all models, except UKESM1, have a bulge with a high model bias around 300-400 hPa, which is at or near the tropopause, implying that most models simulate the tropopause height too low (having larger stratospheric $O_3$ concentrations appearing too low in altitude). This results in a positive bias of about 20% for the MMM around the tropopause. This feature in models was also reported in AMAP (2015), where model biases were particularly large at Ny Alesund and Summit. They associated those with differences in the transport of air masses from the stratosphere. This issue will have an impact on estimating the tropospheric $O_3$ burden, which is a common climate diagnostic (e.g., Griffiths et al., 2021).

At Alert, there are both surface and ozonesonde measurements, and we find that the results in the lowest levels of the Alert ozonesonde comparisons (Figure S.1) are consistent with the model biases found in Figure 8 in that both show the models underestimating winter and fall $O_3$, overestimating spring, and matching well with observations in the summer at this location.

Note that the models' monthly average $O_3$ values were used in this comparison, which does not match the time of day and day of the week as the ozonesonde measurements. However, when a careful time-matching to 3-hourly model output is carried out, the general features of the model biases remain the same (Figure S.2), likely because of the lack of a strong diurnal cycle in Arctic $O_3$ and its relatively long lifetime in the free troposphere.





The results of this model evaluation of the Arctic $O_3$ vertical profiles are consistent with Whaley et al. (2022),
which compared the same model simulations to TES $O_3$ retrievals throughout the troposphere at lower Arctic
locations (~60-70 °N). They found models to be biased low (around -10%), though the TES measurements have
been shown to be biased high by about the same amount (+13% bias in TES measurements reported in Verstraeten
et al., (2013)). They also saw a small positive shift in the model bias profile around 300 hPa as well. Finally, the
Whaley et al. (2022) study included $O_3$, $NO_x$, $CH_4$, and CO comparisons to the Atmosphere Chemistry Experiment
(ACE)-Fourier Transform Spectrometer (FTS) satellite instrument, and those results also implied, independently,
that the modeled tropopause heights are too low.
**6.3 Vertical distribution of $O_3$ precursors**
Intensive field measurement campaigns using aircraft provide the most detailed observational constraint on vertical
profiles of tropospheric $O_3$ precursors in the Arctic. While these datasets tend to provide excellent spatial and
temporal resolution measurements on a wide range of species, they are episodic in nature, often covering only a
period of a few days to several weeks, flying in specific regions of the Arctic and often targeting specific layers or
plumes. For example, Ancellet et al. (2016) examined aircraft, lidar and ozonesonde data over Canada and
Greenland during the summer of 2008 POLARCAT campaigns (Law et al., 2014). This study showed clear
latitudinal and longitudinal variations in the origins of sampled air masses based on back trajectories and $O_3$-
potential vorticity (PV) correlations. While downward transport of $O_3$ was important over Greenland, air masses
with higher $O_3$ were attributed to North American boreal fires over Canada. Transport of polluted air masses from
mid-latitudes also contributed, for example from Asia north of 80 °N.
The airborne NASA Atom (Atmospheric Tomography) mission (Wofsy et al., 2018) has undertaken extensive
surveying of the global troposphere. This includes repeated vertical profile measurements between 60 °N and 90
°N providing useful insights into the variation of $O_3$ and its precursors through the depth of the Arctic troposphere
at different times of the year. Figure 9 shows these mean results and their standard deviation on the left-side panels,
while the equivalent MMM results are on the right-side panels. The models' monthly mean results went into the
MMM calculation and the standard deviation from the models is shown.
The results show that near-surface $NO_2$ is greatly enhanced during winter, associated with a longer $NO_2$ lifetime
and accumulation of pollution in the Arctic haze. The MMM simulates the surface $NO_2$ increase and the seasonality
of the $NO_2$ profiles reasonably well. However, generally, the modeled $NO_2$ is biased low in the tropospheric profile,
having average values of about 15 pptv in the 2-6 km range, whereas the measurements are about 25 pptv on
average. This underestimate is consistent with that found at the surface in Whaley et al. (2022). PAN is also
enhanced at the surface in the winter and can thermally decompose in the spring and summer to release $NO_x$. The
MMM generally overestimates PAN (Figure 9c-d) and does not simulate the same shape in vertical profiles. For
example, models are not able to simulate the wintertime surface level increase in PAN, and they have the inverse
shape in April/May than the observed profile. The best agreement is in summertime PAN (July-Aug), when the
MMM vertical profile better matches that of the observations. The underestimate of $NO_x$ and the lack of winter
surface increases in PAN by the models may be a reason why the wintertime surface $O_3$ concentrations in Section
5.2 and Figure 5 were underestimated. That said, it is possible that the $NO_x$ measurements are biased high and the
PAN measurements are biased low due to thermal degradation in the sample line.



In line with ozonesonde data and previous airborne campaigns (AMAP, 2015), ATom profiles also demonstrate a
springtime enhancement in $O_3$ extending through the troposphere, with evidence of stratospheric influence in the
upper troposphere and lower $O_3$ in the summertime lower troposphere. The models capture that springtime $O_3$
enhancement as well. Summer enhancements in $O_3$ precursors, such as CO and PAN in the mid-troposphere, were
also observed associated with the import of forest fire and anthropogenic emissions from lower latitudes, as also
seen during POLARCAT in 2008. The models capture this feature for PAN, but less so for CO. Indeed, most
models underestimate CO. The annual mean, MMM bias for surface CO in the northern hemisphere has been
reported to be -30% (Whaley et al., 2022). Figure 9 shows that below the tropopause, modeled $O_3$ is actually close
to observed $O_3$, despite the significant MMM biases for CO, $NO_x$, and PAN. Around the tropopause, the aircraft
data show the same issue that the ozonesonde data did – that models overestimate $O_3$ significantly near the
tropopause.

**7. Conclusions**

Recent research on Arctic tropospheric $O_3$ has resulted in improvements to our understanding of this pollutant and
GHG in the rapidly changing and sensitive Arctic environment. We have shown in this study that Arctic surface
$O_3$ seasonal cycles are different depending on whether sites are near the coast, inland, or at high elevation. Coastal
sites have springtime minima due to halogen chemistry causing ODEs and show a maximum during the winter.
The inland, near-Arctic circle locations have quite consistent seasonal cycles, with maxima in April and minima
in August. While the high-elevation sites, less influenced by halogen chemistry than coastal locations, are more
variable; Summit has a later maximum (May), and minimum (September), while Zeppelin has an earlier maximum
(March) and minimum (July).
Despite model development that has occurred since the 2015 AMAP assessment report on ozone (AMAP, 2015)
to add processes, improve parameterizations, increase resolution, etc, the resulting performance of the models
remains more or less the same in terms of model variability and biases compared to measured $O_3$ and $O_3$-precursor
species in the Arctic. Model results for CO would improve if CO emissions from combustion were increased, as
suggested in the literature. It would also be useful to compare modeled OH in the Arctic, but that was beyond the
scope of this study. However, as Arctic $O_3$ is limited by $NO_x$ availability, improvements to CO may not have a
large effect on $O_3$. Improvements to modeled PAN and $NO_x$ are needed, however, sensitivity studies to determine
the cause of the model biases will be required to improve model performance for those species. For surface $O_3$
distributions in the Arctic, models simulate background levels reasonably well (e.g., at the high-elevation location
of Summit), but surface bromine/halogen chemistry needs to be included to simulate springtime surface $O_3$ at
coastal Arctic locations (e.g., Villum, Alert, and Utqiagvik).  Except near the tropopause, models simulate $O_3$
throughout the vertical profile well, with the MMM performing best at +/- 8% depending on the location and
altitude in the troposphere. Attention to improving the height of the modeled tropopause and/or the stratosphere-
tropospheric exchange is still required since downward transport of high stratospheric $O_3$ concentrations is causing
model biases around 6 to 8 km (400 to 300 hPa) to be significantly large (>20%).
While they are logistically challenging, additional $O_3$ measurements in the Arctic, such as $O_3$ deposition
measurements, observations of stratospheric-tropospheric exchange, and $O_3$ concentrations in the Siberian Arctic,
together with long-term measurements of $O_3$ precursors, would be particularly helpful to improve our



understanding and modeling capabilities. This is particularly important as climate change alters the chemistry and
dynamics of tropospheric $O_3$ in the future.
**Author contributions**
CHW, KSL, JLH, HS, SRA, JL, and JBP wrote the manuscript and created Figures 3-9. RYC, JF, and XD provided
the GEOS-Chem model output. JF, ST, and DT edited and provided comments on the manuscript. JHC provided
the DEHM model output. GF, UI, and KT provided the GISS-E2.1 model output. MG and ST provided the EMEP-
MSCW model output. KSL, JCR, TO, and LM provided the WRF-Chem model output. MD and NO provided the
MRI-ESM2 model output. DAP provided the CMAM model output. LP provided the CESM model output. RS
provided the OsloCTM model output. MAT provided the MATCH-SALSA model output. SRA and STT provided
the UKESM1 model output. DT provided the Canadian ozonesonde measurements, and DW provided Alert
datasets. MF and KvS provided the model strategy for this project.
**Competing interests**
At least one of the (co-)authors is a member of the editorial board of Atmospheric Chemistry and Physics. The
peer-review process was guided by an independent editor, and the authors also have no other competing interests
to declare.
**Special issue statement**
This article is part of the special issue "Arctic climate, air quality, and health impacts from short-lived climate
forcers (SLCFs): contributions from the AMAP Expert Group (ACP/BG inter-journal SI)". It is not associated
with a conference.
**Acknowledgements**
We wish to acknowledge Wang and Pratt for their figure originally published in PNAS, as well as Seabrooke and
Whiteway for their figure originally published in JGR. We thank Garance Bergeron for providing the processed
ATOM observations. The technicians and logistical support staff at the different stations are gratefully
acknowledged for their work.
**Financial support**
Makoto Deushi and Naga Oshima were supported by the Japan Society for the Promotion of Science KAKENHI
(grant numbers: JP18H03363, JP18H05292, JP19K12312, JP20K04070 and JP21H03582), the Environment
Research and Technology Development Fund (JPMEERF20202003 and JPMEERF20205001) of the
Environmental Restoration and Conservation Agency of Japan, the Arctic Challenge for Sustainability II (ArCS
II), Program Grant Number JPMXD1420318865, and a grant for the Global Environmental Research Coordination
System from the Ministry of the Environment, Japan (MLIT1753 and MLIT2253). Joakim Langner and Manu A.
Thomas were supported by the Swedish Environmental Protection Agency through contracts NV-03174-20 and
the Swedish Clean Air and Climate research program. Svetlana Tsyro and Michael Gauss have received support
from the AMAP Secretariat and the EMEP Trust Fund. Ulas Im received support from the Aarhus University
Interdisciplinary Centre for Climate Change (iClimate) OH fund (no. 2020-0162731), the FREYA project funded
by the Nordic Council of Ministers (grant agreement nos. MST-227-00036 and MFVM-2019-13476), and the
EVAM-SLCF funded by the Danish Environmental Agency (grant agreement no. MST-112- 00298). Henrik Skov
received funding from the Danish Ministry for Energy, Climate and Utilities (Grant agreement no. 2018-3767)
and Danish Environmental Agency (grant agreement no. MST-113- 00140) and AMA. Kostas Tsigaridis and


Gregory Faluvegi received support from the NASA Modeling, Analysis and Prediction Program (MAP). Steven
T Turnock would also like to acknowledge the financial support received from the Arctic Monitoring and
Assessment Programme. Kathy S. Law, Jean-Christophe Raut, Louis Marelle and Tatsuo Onishi (LATMOS)
acknowledge support from EU iCUPE (Integrating and Comprehensive Understanding on Polar Environments)
project (grant agreement n°689443), under the European Network for Observing our Changing Planet (ERA-
Planet), and from access to IDRIS HPC resources (GENCI allocation A009017141) and the IPSL mesoscale
computing center (CICLAD: Calcul Intensif pour le CLimat, l'Atmosphère et la Dynamique) for model
simulations. Jesper Christensen (DEHM model) acknowledges Danish Environmental Protection Agency and
Danish Energy Agency (DANCEA funds for Environmental Support to the Arctic Region project: grant no. 2019-
7975, grant no. MST-112- 00298, grant no. TAS 4005-0153). Stephen R. Arnold and Steven T. Turnock both
acknowledge the financial support received from the Arctic Monitoring and Assessment Programme. Stephen R.
Arnold also acknowledges support from the UK Natural Environment Research Council and Belmont Forum via
the ACRoBEAR project (grant NE/T013672/1). Joshua Fu received funding from the Oak Ridge Leadership
Computing Facility at the Oak Ridge National Laboratory, which is supported by the Office of Science of the U.S.
Department of Energy under Contract No. DE-AC05-00OR22725.

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

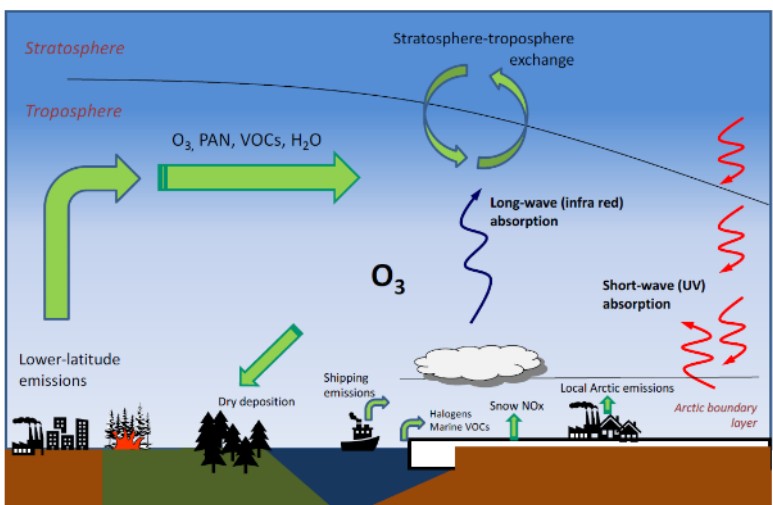


**Figure 1:** Schematic of Arctic tropospheric $O_3$ sources, sinks, and relevant processes.

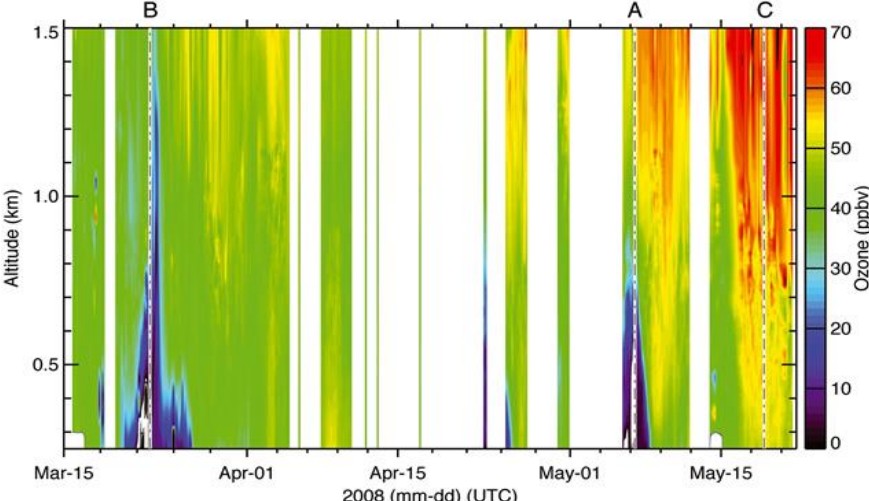


**Figure 2:** Ozone lidar measurements from Eureka in the spring of 2008 showing effects of large-scale
meteorology including low $O_3$ in the lower troposphere when air masses originate from the north over the Arctic
Ocean and enhanced $O_3$ during downward transport into the Arctic boundary layer when the airflow was from
the south over mountains. From Figure 3 in Seabrook and Whiteway (2016), JGR Atmospheres, vol. 121, Issue:
4, Pages: 1935-1942, First published: 04 February 2016, DOI: (10.1002/2015JD024114)

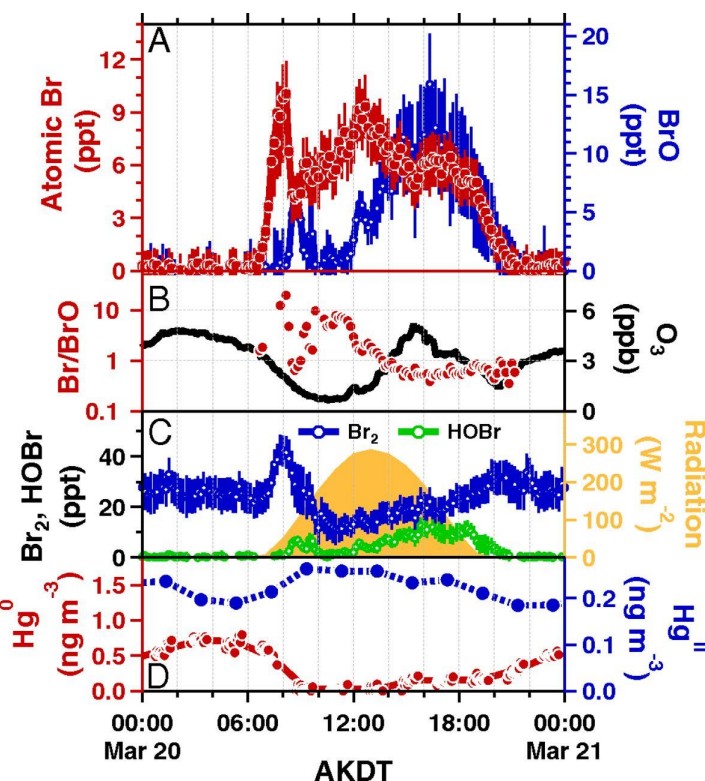


**Figure 3:** Time series at Uqiagvik on 20 March 2012 of measured (A) atomic bromine (Br) and bromine

monoxide (BrO), (B) Br/BrO ratios and $O_3$. Error bars represent propagated measurement uncertainties. Figure

based on Wang et al. (2019, PNAS). (EPS figure provided for the report). From Figure 2 in Wang et al. (2019),

PNAS, vol. 116, no. 29, pages 14479-14484, Copyright (2019) National Academy of Sciences.






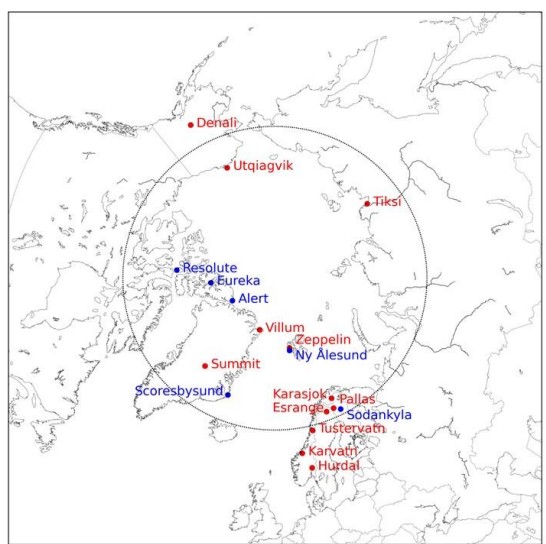

| Name | Longitude | Latitude | Elevation (msl) |
|---|---|---|---|
| Alert | -62.33 | 82.50 | 66 |
| Denali | -151.19 | 63.11 | 660 |
| Esrange | 21.07 | 67.88 | 475 |
| Eureka | -86.4 | 80.1 | 83 |
| Hurdal | 11.07 | 60.44 | 300 |
| Karasjok | 25.22 | 69.47 | 333 |
| Karvatn | 8.88 | 62.78 | 210 |
| Pallas | 24.25 | 67.97 | 565 |
| Summit | -38.48 | 72.57 | 3211 |
| Tiksi | 128.9 | 71.6 | 10 |
| Tustervatn | 13.87 | 65.83 | 439 |
| Utqiagvik | -156.61 | 71.32 | 11 |
| Villum | -16.67 | 81.60 | 20 |
| Zeppelin | 11.89 | 78.91 | 474 |
| Ny Ålesund | 11.49 | 78.51 | |
| Resolute | -94.82 | 74.68 | |
| Scoresbysund | -21.97 | 70.48 | |
| Sodankyla | 26.65 | 67.37 | |


**Figure 4.** Map of the surface (red) and ozonesonde (blue) sites, cited in the present study, with coordinates and

elevation. Eureka and Alert are both surface and soundings sites. 'Utqiagvik' was formerly called 'Barrow'.

The Arctic Circle at 66.55° N is also shown in the figure for reference.

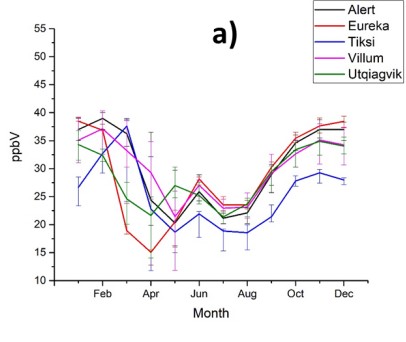

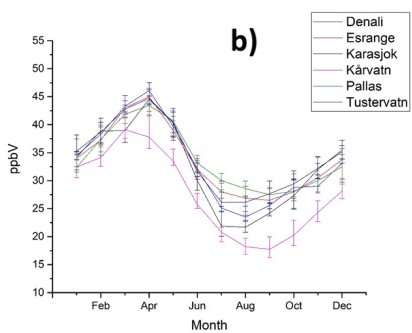

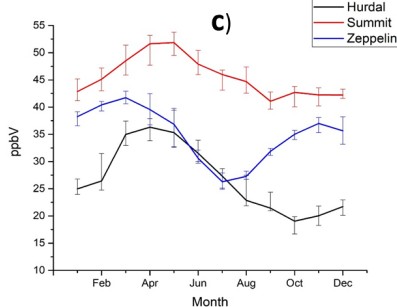

**Figure 5:** Seasonal behavior of surface O$_3$ at selected Arctic stations that are representative of a) coastal high
Arctic b) near Arctic Circle and c) inland and high-elevation sites. Monthly medians are calculated for the period
2003 to 2018. Data were not available from 2003 to 2006 for Villum and 2004 and 2013-2015 for Alert. The
error bars show upper (75%) and lower (25%) quartiles.




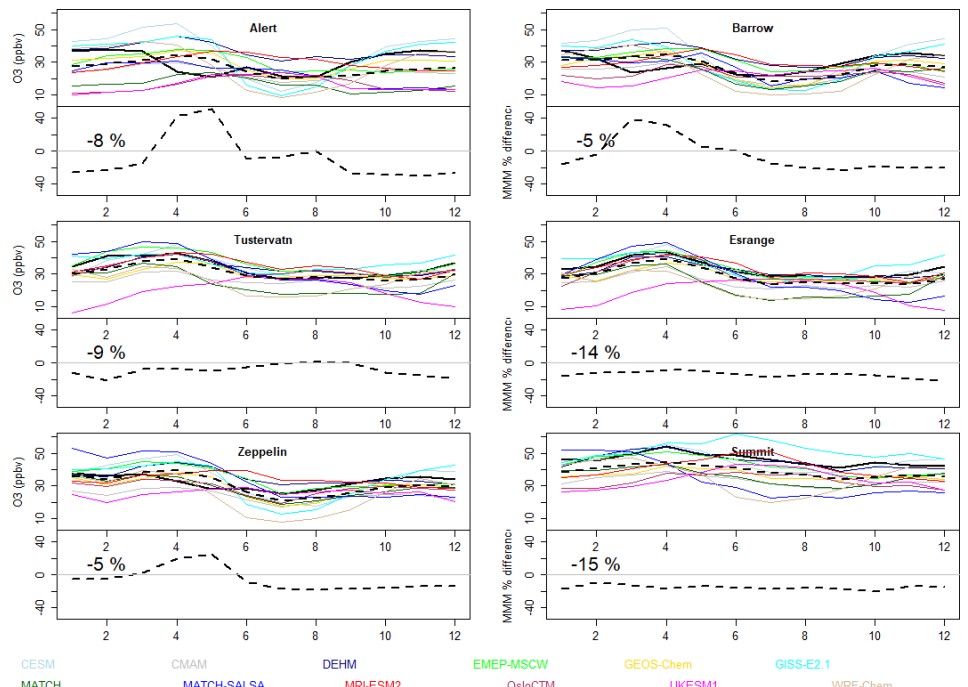


**Figure 6:** Arctic surface O$_3$ by month; seasonal cycle model comparisons. Top row: coastal high-Arctic sites;

middle row: near-Arctic circle sites; bottom row: high elevation sites. The solid black line is the observed O$_3$

monthly means, and the dashed black line is the multi-model median. Bottom row: sub-panels show the MMM

% difference [(MMM - measurements)/measurements*100].

*Note model results are from the 2014-15 mean. When available, the same years are used for the observations.

However, Alert did not have data for 2014-15, so its most recent years were used: 2010-2013. Summit had 2014,

but only 1 month in 2015, so its 2013-2015 data were used.

1134

1135

1136





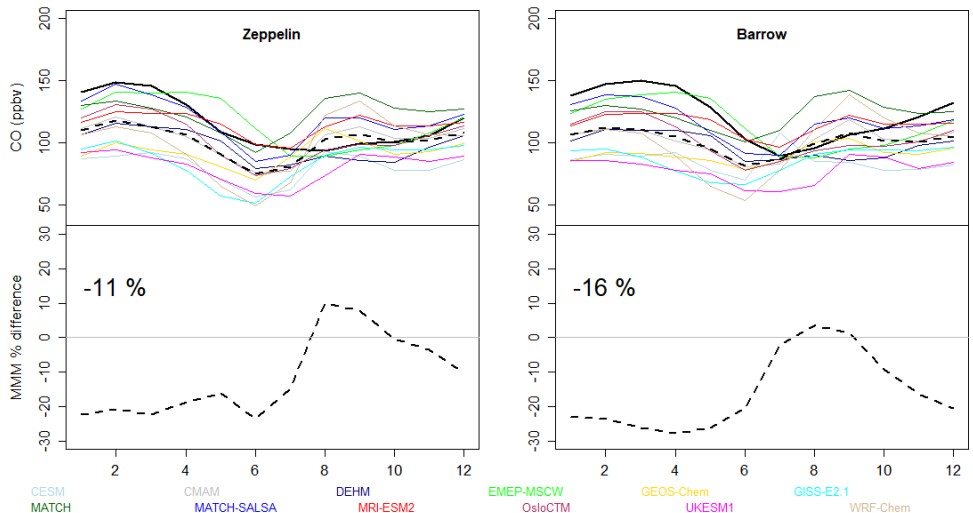

**Figure 7:** Arctic surface CO by month; seasonal cycle model comparisons. The solid black line is the observed

CO monthly means, and the dashed black line is the multi-model median (MMM). Bottom panels show the

MMM % difference [(MMM - measurements)/measurements*100].

*Note model results are from the 2014-15 mean. When available, the same years are used for the observations.

However, for Zeppelin observations are the mean of 2013-14.



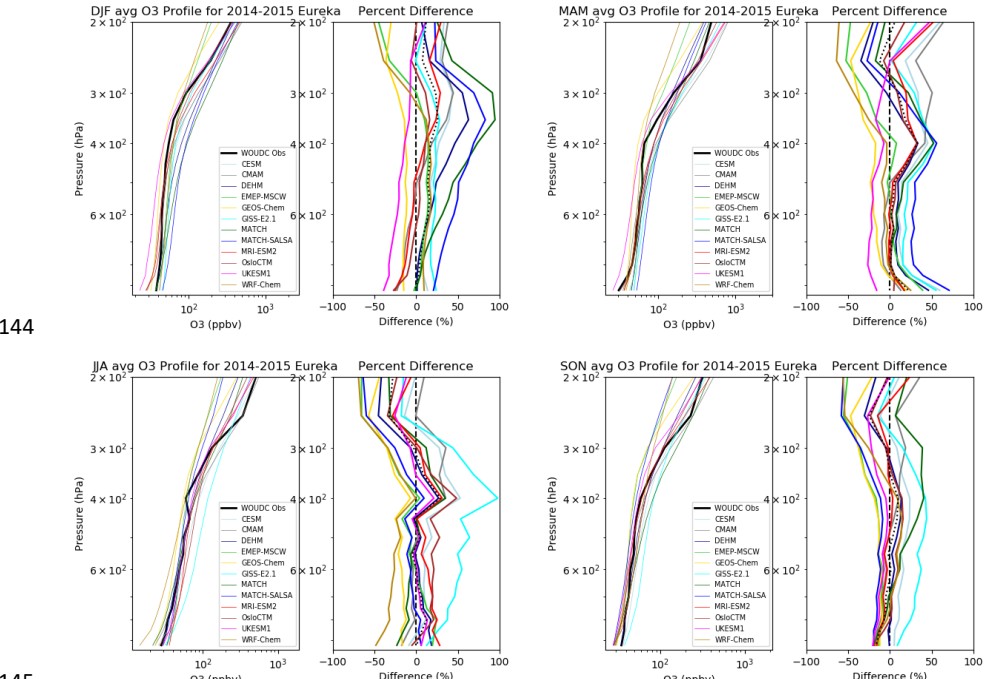

**Figure 8**: Comparison between observed (thick black line on left panels) and AMAP models' (colored lines) $O_3$ seasonal averages for 2014-15 at Eureka, NV, Canada. These use monthly mean model output. On each right panel, the dotted black line is the MMM, and the dashed black line shows zero bias for reference. See supplement (Figure S.1) for the rest of the ozonesonde locations, and a sample comparison done with 3-hourly model output (Figure S.2).



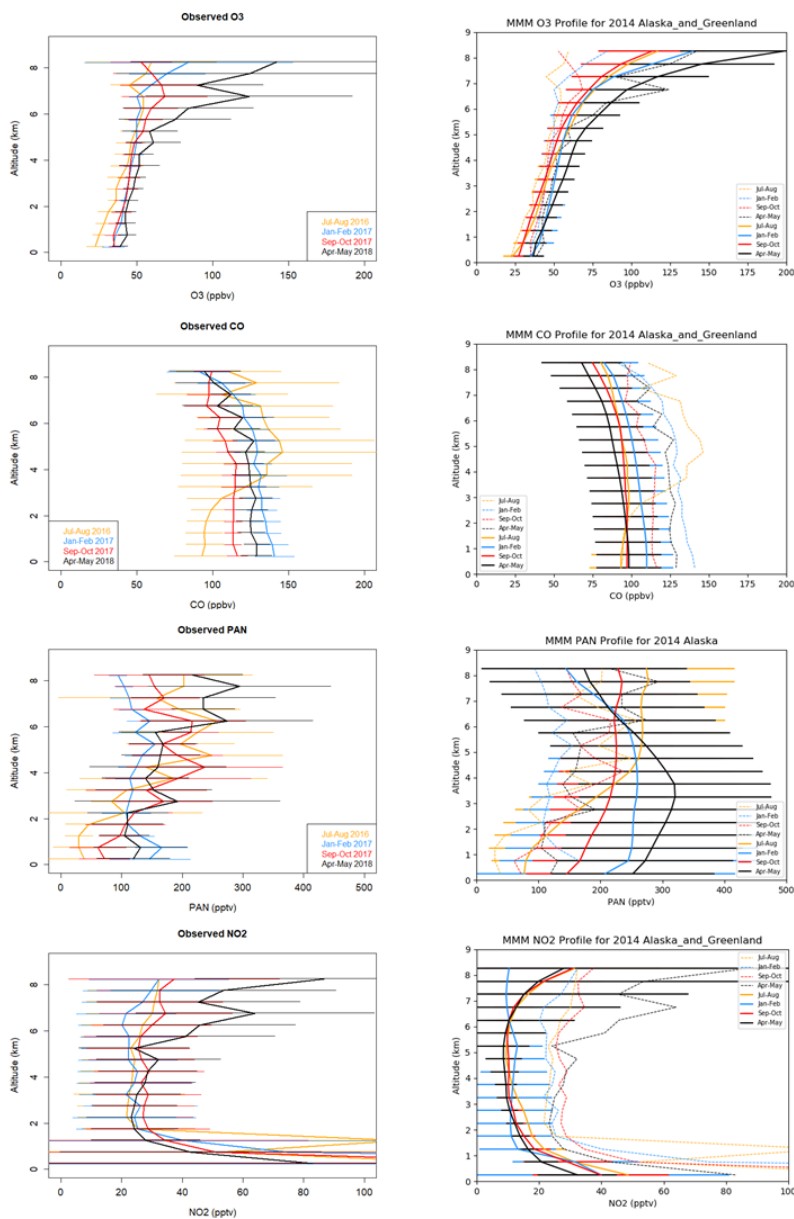


**Figure 9:** Mean vertical profiles of O$_3$, CO, PAN and NO$_2$ **(left)** measured in Alaska and Greenland from the
NASA ATom missions during summer 2016, winter 2017, autumn 2017 and spring 2018 (horizontal lines
indicate 1 standard deviation spread around mean values at each altitude. **(right)** the MMM for the years 2014-
15 (with the MMM standard deviation as horizontal lines). The observations appear as dashed lines in the right
panels, for ease of comparing to the MMM.