# Peer review of "Arctic tropospheric ozone: assessment of current knowledge and model performance"

_Atmospheric Chemistry and Physics, 2022_

## Author Response (AR1)

Response to reviewers
*Reviewers comments in italic*
Authors' responses start with [AR]

Reviewer 1

*Main comments to the paper*

*This is a well written and comprehensive paper assessing and reviewing the present state of knowledge of ozone in the Arctic troposphere. Despite that, this reviewer asks for major changes before the paper is accepted for publication. The reason for this view is given in the following. The authors make many shortcuts when summarising the results from existing publications. It seems as the authors are so focussed on trying to make an overall review that they leave out all the details and present everything as generally valid for the Arctic troposphere. The authors should investigate more closely the publications they refer to and clarify what conclusions are valid for certain sites and time periods only and to what extent the findings are relevant for the Arctic in general. The paper should provide more details of the previous work and should be less conclusive regarding the Arctic as a whole.*

[AR] Thank you for your thorough and helpful review of our paper. In our revision, we have addressed all of your comments, adding in additional information, clarifications, and references, and removing some when appropriate. While we realise the impression that the paper gives off, we did not intend for it to be a literature review paper, so in some cases, we have shortened Sections 2 and 4 in the revision. Please see below for specific responses to your concerns.

*Specific comments to the content*

*L122 With a reference to Walker et al. (2012) the authors claim that during summer, the dominant source of Arctic tropospheric O3 is in-situ production in the Arctic, which in July is said to contribute more than 50% of O3 in the Arctic boundary layer. This number seems high, and the authors are asked to provide more information what internal Arctic sources this includes as well as the uncertainty of this estimate. It seems like the Walker et al (2012) ref is based on a global model simulation with a coarse 4x5i° resolution and on an adjoint model calculation for a few weeks in 2006 for Alert only. To what extent could these findings be generalised to the whole Arctic as the authors do in the paper? This relates also to the statement in L145 regarding contribution from shipping to surface O3 in the Arctic in summer. The Marelle et al. (2018) paper states that "Ozone production from shipping emissions could be overestimated in the present study, since this is a known artefact of models run at lower resolutions".*

[AR] the Walker et al paper itself does make the same overly general claim (e.g. in the Abstract), which may not be explicitly justified. We do state explicitly that their model results are for July, and we have reworded to make this clear. The dominance for in-situ production in the Arctic is only actually shown in the paper for profiles over Eureka and Ny-Ålesund where it accounts for more than 50% at the surface. The model results do show net production on average in the Arctic north of 60N, so in the revised manuscript, we have softened the text and removed the claim of "dominant in the Arctic". Our statements about the shipping source of emissions were supported by more references than just the Marelle et al. (2018) paper, however, we have removed some text related to that reference in our revision as drivers of future Arctic ozone are less relevant to our current study. Thank you for bringing this to our attention.

*L148 The authors write: «Tuccella et al. (2017) showed that background O3 is influenced by emissions downwind of oil and gas extraction platforms in the southern Norwegian Sea." This is correct, but the authors don't provide any details, and thereby give the impression that this conclusion is generally valid for the whole Arctic boundary layer. In reality, the statement by Tucella et al. (2017) was based on two summer days in July 2012 and only for rather small domains. Furthermore, the ozone impact for the two small domains at these two days were less then 1 ppb on average. Thus, the authors should provide more details and rewrite their statement so that it does not give the impression being a general fact for the whole Arctic.*

[AR] For the Tuccella reference we stated clearly that this was a regional result for the Norwegian Coast and it was given as an example of "progress in improving knowledge of local $O_3$ precursor sources". This is an example of where ozone is already being influenced by local Arctic emissions. Now that this paragraph is shorter overall, we hope that this is clearer in the revision.

*L170 "Interestingly, gradient studies at Barrow showed a positive gradient with height during O3 depletion events (ODE) and atmospheric mercury depletion events (AMDE) suggesting that O3 was removed at the surface due to fast photochemical reactions at or close to snow surfaces initiated by the release of halogen species (Skov et al., 2006)." The link between ODE and AMDE has been extensively studied at Zeppelin/Ny-Ålesund. Why is this not mentioned?*

[AR] Thank you. We have added that these have been observed at Zeppelin as well, and included new references (e.g., Solberg et al, 1996; Berg et al, 2003).

*L211-L212 "… simulations of the years 2014-2015 for comparisons to observations." The authors should provide an explanation why they chose this early period while more recent measurement data are available.*

[AR] While more recent measurement data are available, the emissions that go into model simulations take a long time (many years) to develop. For this reason, it is common for model years to lag behind the most recent measurements. The emissions that we used in this study (ECLIPSEv6b) were available up to 2015 for the historical period, with later years as emissions projections under different future scenarios. We have added this explanation to the revised manuscript.

*L299 "In the high Arctic, there is very little diurnal variation in surface O3, most likely because the local and regional photochemistry is of limited importance most of the time and due to the 24-hour daylight during Arctic spring, summer and Autumn as well as the polar night during winter" This is incomplete. The main driver of diurnal variation in surface ozone in other regions is the deposition to the ground and uptake in vegetation. Due to the inefficient deposition to ice/snow/water surfaces and the sparsity of vegetation, there is very little diurnal cycle in O3 in the Arctic. This is the most important factor. The role of local photochemistry would anyway be very small in the Arctic due to the low levels of NOx and other precursors.*

[AR] Thank you, we have added text related to the inefficient $O_3$ deposition to this discussion about the lack of diurnal cycle in the revised manuscript.

*L320 "Regarding the more southerly Arctic and near-Arctic sites, a latitudinal gradient has been observed in the timing of the spring O3 maximum. Anderson et al. (2017) found that monthly mean observed near-surface O3 concentrations at background sites in Sweden from 1990 to 2013 had a maximum in spring, but the most northerly stations experienced their maximum in April while the more southerly ones in May." Couldn't this simply be explained by the southerly sites being exposed more frequently to polluted air masses from the European continent in May? The phrase "near Arctic sites" is somewhat meaningless. The paper from*

*Anderson et al. (2017) analysed data from all Swedish sites which includes stations down to nearly 55ï°N. It's not clear why these data are included in an assessment study of the Arctic troposphere.*

[AR] We mention in this paragraph that "Seasonal cycles at Arctic stations are not extensively discussed in the literature, …", which is why we wanted to reference the Ansderson et al (2017) paper, as one of the few that did. That paper had northerly, central, and southerly parts of Sweden. We find that their discussion is useful for understanding the difference between Arctic and non-Arctic sites. However, we have modified the text to remove any misleading information, and to use the term "non-Arctic" instead of "near Arctic".

*L325 "In order to get an overview of the annual O3 cycles at different types of Arctic surface measurement sites, we have calculated the monthly medians and interquartile range for the period 2003-2019 for a series of sites. A map of the stations as well as their coordinates and elevation can be seen in Figure 4. Figure 5 illustrates the range of seasonal cycle behaviour observed in the Arctic at different measurement sites." Several of the sites in these figures are surely not Arctic. The Arctic circle is an easy boundary to use for defining the Arctic region but less relevant for atmospheric studies. The conditions are highly different in the eastern/European region compared to the American sector. Besides, several sites in these figures are located much further south than the Arctic circle. The authors should outline this and explain why these sites are relevant for an assessment of the Arctic.*

[AR] Our initial thinking was to have sub-Arctic sites like Karvatn and Hurdal to demonstrate the difference between O3 at those locations versus O3 at firmly Arctic locations. However, at your suggestion, we have removed Karvatn and Hurdal from Figures 4 and 5 and from the discussion to better focus the paper on the Arctic region.

*L341 "The largest differences between the stations are mainly found during the summer months, most likely due to the influence of local sources on photochemical O3 production." I disagree. The local photochemical O3 production at the southerly sites are likely very small. The reason for the differences is most likely due to the distance to the main emission areas in Europe and the frequency of episodes transporting ozone from these areas.*

[AR] Thank you. We have added varying distance to sources to the text in the revised manuscript, along with the Anderson et al (2017) reference, which supports that idea. We have also clarified that some sites may be influenced by shipping, adding the Marelle et al (2016) reference to further strengthen that statement.

*L344 "Kårvatn in Norway has an unusual behavior with an O3 maximum in March, possibly due to the local conditions: The site often shows a pronounced diurnal cycle in O3 due to the location at the bottom of a valley that causes strong inversions leading to an enhanced impact of dry deposition at night on surface O3 (Aas et al., 2017)." Why use this old reference when there exist several newer annual reports? And why use the Kårvatn site at all? It has very little relevance for assessing the Arctic.*

[AR] We have removed Karvatn from the paper, in order to better focus on the Arctic region.

*L347 "Hurdal in Norway is included as an example of a more southerly Scandinavian non-Arctic station, which has an annual variation with a minimum in October while the more northerly stations have minima between July and September (Figure 5c), this difference may be explained by a stronger influence of local air pollution at Hurdal. At Hurdal, winter O3 concentrations are particularly low, probably also in this case due to the influence of local emissions which in this period leads to the removal of O3 by the reaction with NO." What is the relevance of this station for the assessment of the Arctic? If the reason for the ozone*

*differences is explained by local conditions at Hurdal, it seems meaningless to include this site in a paper assessing the Arctic troposphere.*

[AR] We have removed Hurdal from the paper, in order to better focus on the Arctic region.

*L363 "… the occasional ODE has been reported there by Lehrer et al. (1997) and Ianniello et al. (2021)." The Ianniello et al. (2021) paper was based on measurements down at the coast in Ny-Ålesund (40 masl) and not on the Zeppelin Mountain. Either this ref should be removed, or it should be stated clearly that these findings refer to a coastal location. And why aren't more papers from Zeppelin mentioned? The occurrence of ODEs at Zeppelin has been extensively documented but none of these studies are mentioned in the present manuscript.*

[AR] Thank you. We have corrected these statements, and also added several references for ODEs observed on Zeppelin mountain.

*L424-440 There have been long-term measurements of VOCs at both Zeppelin and Pallas, so why are these data not mentioned? Presently, the assessment of VOCs in this paper is insufficient and should be significantly extended.*

[AR] While VOCs are important $O_3$ precursors generally, we mention in section 5.3 (first sentence of the 3rd paragraph) that in the NOx-limited, remote Arctic, the low concentrations of VOCs don't have a strong impact on $O_3$ there. There are also many VOC species, and evaluating them all could be a whole paper in and of itself. Finally, only a couple of models provided VOC output (e.g., $C_2H_6$), and only as monthly means. It is common for some climate models to simulate only grouped VOCs. Given the model dataset that this paper is written about, it means that we cannot delve into the VOC topic with sufficient thoroughness, nor as we mention above would the VOCs necessarily have a significant impact on $O_3$ concentrations.
While there are not many VOC measurements in the Arctic, this paragraph cites a few VOC studies and measurements there, and in our revised paper, we have also added references for the long term VOC measurements in Zeppelin and Pallas (e.g., Platt et al, 2022 and Hellén et al, 2015, respectively). Thank you for bringing those to our attention.
We have also added text in the conclusion to suggest that it as a topic for future work.

*L462 "Methane has more than doubled since preindustrial times (from 0.8 ppmv to 1.8 ppmv)" What year is this based on? And is it referring to the annual mean? Presently, methane levels have exceeded 2 ppmv in the Arctic.*

[AR] Apologies for the lack of clarity around these values. These represent global mean mixing ratios. Detailed estimates from the IPCC AR6 report indicate that global mean mixing ratio of methane was ~1870 ppb in 2019, an increase of ~1140 ppb since 1750. We have reworded the sentence and added the reference to make this clear.

*L495 "All of these stations are located in the sector 95°W to 27°E meaning that regular soundings are lacking in a large sector of the Arctic." Technically speaking, this is correct, but longitude values are difficult to imagine in polar regions, so it is recommended that the authors rephrase this sentence. In practice, the most obvious lack of ozone sonde data is from the Russian sector + from Alaska.*

[AR] Thank you. We have reworded this in the revised manuscript to specify ozonesondes are mainly in Canada and Europe, and lacking in Russia and Alaska.

*L595 "…surface bromine/halogen chemistry needs to be included to simulate springtime surface $O_3$ at coastal Arctic locations (e.g., Villum, Alert, and Utqiagvik)." Although the effect*

*of the halogen chemistry is most visible at coastal station, this chemistry is also needed for a proper modelling of ozone elsewhere in the Arctic due to advection of the ODEs.*

[AR] Agreed, we have reworded this in the revised manuscript to be clearer about that.

*Fig 5 What is the rationale for grouping together so different locations as Hurdal, Zeppelin and Summit in this plot? This seems very strange and e.g., Hurdal has no relevance for assessing Arctic ozone levels. It should also be mentioned that ozone monitoring at Karasjok ended in 2010.*

[AR] We have removed Hurdal from the paper entirely to better focus on the Arctic region. The remaining locations in Figure 5c are Zeppelin mountain and Summit, which, while quite different, both qualify as high elevation sites. We have added some text regarding Karasjok's time series. Thank you for the suggestions.

*General comment regarding data use and acknowledgement*

*The paper contains a vast number of references to previous publications, but there is very little information of how the data presented in this paper have been collected. This regards not only technical issues such as web addresses etc, but it also seems that the contribution from the various measurement data providers and institutions to this paper are absent. This represents a common modeller's attitude to the science: Open-source data could apparently just be downloaded and used without acknowledging the years of experience at the data providers' institutions. Some data from the Finnish site Pallas are included, but why are there no co-authors from FMI? And why are there no mentioning of the long time series of VOC measurements from this site? Ozone data from Esrange in Northern Sweden are included, but no acknowledgement to the data provider IVL is given and no IVL personnel are included as co-authors. The same holds for measurement data from Norway: There is no acknowledgement to NILU, and no NILU people are included as co-authors. Just a reference to an old monitoring report is given: Aas et al., 2017. And why are the VOC measurement data from Zeppelin not mentioned?*

[AR]: Thank you for raising these points. The revised manuscript now has a "data and code availability" section providing sources for the model and measurement data. While Henrik Skov and David Tarasick (both measurement specialists) are co-authors on this paper, we have now also contacted the rest of the relevant measurement scientists and offered them co-authorship. The revised manuscript now has 10 new authors and 3 new acknowledgements. We apologise that our original manuscript overlooked these important contributors.

[AR] Regarding VOC measurements, we have added text in the revised manuscript to mention them being measured at Zeppelin and Pallas, but as mentioned above, there are (a) many VOC species and it is difficult to choose which ones to evaluate, (b) very few Arctic VOC measurements overall, so we wouldn't want to presume a few locations represent the whole Arctic, and (c) most models did not provide VOC output, other than CO (and CH4) – for all of these reasons, evaluating VOCs in the Arctic is beyond the scope of this paper. But we've added text to clarify this (see AR above) and suggest that it is important for further/future work.

Reviewer 2

*In this study, Whaley and coauthors perform an ambitious comparison between simulations and observation of tropospheric ozone in the Arctic. The model output result from simulations for the Arctic Monitoring and Assessment Programme (AMAP) performed with a number of models with different characteristics. The comparison also includes ozone precursors like NOx*

*and CO. The results for the different compounds are evaluated with surface, aircraft, and satellite measurements. The major conclusion is that the performance of the models concerning the correct simulation of tropospheric ozone has not significantly increased in the last 10 to 20 years. Overall, this is an interesting paper, which merits publication in ACP. However, similar to referee #1 I recommend major revisions before publication.*

[AR] Thank you for your thorough and helpful review of our paper. In our revision, we have addressed all of your comments. Please see below for specific responses to your concerns.

*The manuscript appears like an offspring of a previously published paper on a model evaluation of short-lived climate forcers in the Arctic (Whaley et al., 2022). This paper also included model results for ozone in the Arctic including a comparison to observations. This manuscript here describes the results and comparisons for ozone in more detail. However, as already mentioned by referee #1 a number of relevant processes and studies related to arctic ozone are mentioned, but are not analyzed in sufficient detail. This concerns for example the springtime depletion of ozone and the related halogen chemistry, for which a number of requirements have been postulated in the literature (temperature thresholds, presence of first-year ice, second-year ice, or frost flowers, etc.) or the emission of NOx from the snowpack as local source. Moreover, previous reviews and modeling studies as well as observations beyond the coastal stations are missing. Some examples are given in the comments below.*

[AR] Thank you for suggesting additional information for this paper. In our revision, we have added additional information, clarifications, and references, and removed some when appropriate. While we realise the impression that the paper gives off, we did not intend for it to be a literature review paper, so in some cases, we have shortened Sections 2 and 4 in the revision. Please see below for specific responses to your concerns.

*Further comments*

*The authors may want to add a point on iodine chemistry, which was very recently pointed out as important for the destruction of Arctic ozone (Benavent, N., et al., Substantial contribution of iodine to Arctic ozone destruction, Nature Geoscience, https://doi.org /10.1038/s41561-022-01018-w, 2022).*

[AR] Thank you for this reference. In the revised manuscript we have added this in Section 2.2 for ozone sinks.

*Fig. 1: I'm somewhat confused by the representation of the Arctic. The Arctic is actually a large ocean covered permanently or seasonally by sea ice with continents around. While there are some local emissions due to shipping or other marine activities in the Arctic Ocean, the majority of the emissions are on the continents around the Arctic Ocean. I think this could be better represented in the figure. The figure also gives the impression that the stratosphere-troposphere exchange happens in the Arctic (and only in the Arctic). Isn't there also a fraction of Arctic ozone with a stratospheric origin that is mixed down in lower latitudes and then it is transported to the Arctic in the free troposphere. Is this correct or is this negligible? If this plays a role, I think this could be added to the figure, too.*

[AR] Thank you. While Figure 1 is meant to be an uncluttered schematic and not fully geographically accurate, we have revised the figure based on some of these important points you've raised.

*Finally, in L. 104f it is stated "…which show marked seasonal and inter-annual variations (Figure 1)." It is not clear to me, how these variations are represented in the figure.*

[AR] Thank you - we have moved the reference to figure 1 to earlier in the sentence when the different processes are listed.

*L.148: "…shipping would become the main surface O3 precursor source."?*

[AR] Yes, thank you for pointing that out. Due to a comment from reviewer 1, we have actually removed this sentence in the revision.

*L. 178f: "These phenomena are most commonly observed at Arctic coastal locations in March/April…" I'm missing here the reference to Helmig et al., who did the first review on Arctic ozone observations (Helmig, D., et al., A review of surface ozone in the polar regions, Atmos.Environ. 41, 5138-5161, 2007.) Moreover, it would be important to mention here that the few available springtime O3 measurements over the Arctic Ocean actually show an even more pronounced depletion (Bottenheim, J.W., et al., Ozone in the boundary layer air over the Arctic Ocean: Measurements during the TARA transpolar drift 2006-2008, Atmos.Chem.Phys. 9, 4545-4557, 2009; Jacobi, H.-W., et al., Observation of widespread depletion of ozone in the springtime boundary layer of the Central Arctic linked to mesoscale synoptic conditions, J.Geophys.Res. 115, D17302, doi: 10.1029/2010JD013940, 2010 and the more recent observations during MOSAiC also in Benavent, N., et al., Substantial contribution of iodine to Arctic ozone destruction, Nature Geosci., https://doi.org/10.1038/s41561-022-01018-w, 2022)*

[AR] Thank you for mentioning these important references, which have been added to the revised manuscript.

*L. 181ff: "Interestingly, Yang et al. (2020) and Huang et al. (2020) were able to explain major depletion events in a model study by introducing the wind-induced release of bromine from the snowpack. However, the models could not explain the depletion events observed at low wind speeds. Swanson et al. (2022) used the GEOS-Chem model to show that both blowing snow and the snowpack are important sources of bromine during the spring." It is unclear why these three studies are selected here, while many more modeling attempts can be found in the literature, e.g. (without being complete) Toyota, K., et al., Analysis of reactive bromine production and ozone depletion in the Arctic boundary layer using 3-D simulation with the GEM-AQ: Inference from synoptic-scale patterns, Atmos.Chem.Phys. 11, 3949-3979, 2011; Yang, X., et al., Snow-sourced bromine and its implications for polar tropospheric ozone, Atmos. Chem. Phys., 10, 7763–7773, https://doi.org/10.5194/acp-10- 7763-2010, 2010.*

[AR] Thank you for mentioning these important references as well. We've added them to the revised manuscript.

*L. 192f: "Their findings suggest a dark wintertime source of reactive bromine (halogens) that could feed halogen photochemistry at lower latitudes as the sun returns." A direct impact of a dark mechanism on ozone (and mercury) has been observed over sea ice in Antarctica according to Nerentorp Mastromonaco, M., et al., Antarctic winter mercury and ozone depletion events over sea ice, Atmos.Environ. 129, 125-132, doi: 10.1016/j.atmosenv. 2016.01.023, 2016.*

[AR] Thank you for suggesting this reference to support this topic. We have added it to the revised manuscript.

*Ch. 3: For the comparison of the model output with the observations the authors use a multi-model median using the model grid boxes containing the measurement site. However, the models have different spatial resolution. Is this considered? Does this impact the calculated medians?*

[AR] As noted in Section 3, the models do have different spatial resolution, however, we have still used the nearest model grid box to compare to the measurement location. Under different circumstances when there are many measurements that fall within a model grid box, those measurements would first be averaged together and compared to the model output in that gridbox. However, in the Arctic, the measurements were sparse, so there is no averaging, and no special treatment of the multi-model median. We mention at the end of Section 3 our methodology for the comparisons, with a small revision for clarity.

*L. 254f: "…but negative IASI biases were found compared with aircraft data in the lower troposphere, due to low thermal contrast in the Arctic boundary layer." Does that mean that there is a bias in the retrieval of O3 from IASI data in the boundary layer or do the authors refer to a boundary layer process? Please clarify.*

[AR] Apologies that this sentence was not clear. We have re-written this as:
"... but low thermal contrast between the Arctic surface and boundary layer was found to produce bias in IASI retrievals compared with aircraft measurements in the Arctic lower troposphere."

*Ch. 5: The authors refer here for the first time to the "high Arctic". It would be good to define earlier in the manuscript what the authors mean by this region and probably also if they refer to the Arctic itself.*

[AR] Thank you for this suggestion. We have added a clarification that we mean the observation sites located >70°N as "high Arctic" (those in Fig 5a), to contrast them with the observation sites closer to the Arctic circle at 66.5°N (those in Fig 5b)

*L. 303: Figure 5a appears in the text before Figure 4.*

[AR] In the revised manuscript we make sure to reference the figures in order.

*L. 317: "Seasonal cycles at Arctic stations are not extensively discussed in the literature,…" See Helmig et al., 2007 (see above).*

[AR] Thank you. We have added that reference and revised the sentence.

*L. 369ff: "Therefore, while none of the Arctic sites currently exhibit summertime surface maxima due to photochemical production, as often observed in polluted locations further south, this may change in the future with increasing local anthropogenic emissions (e.g. Marelle et al.2018)." This is highly speculative since at the moment summertime minima are still observed. Maybe the authors could give a more quantitative estimate here, i.e. by how much would the different O3 precursors need to grow to actually turn the Arctic into an area with summertime photochemical production of ozone?*

[AR] Results from the same model experiment are also presented in Law et al (2017, Figure 4), showing 35 ppbv of ozone in Utqiagvik in July with additional Arctic diversion shipping emissions of +150 kT(NOx)/month. This is comparable to the current annual max O3 at Utqiagvik/Barrow shown on Figure 6. The older modelling study of Granier et al (2006, Figure 2) found 50 ppbv of ozone in July at Utqiagvik with +560 kT/month of NOx. We have added these references to provide more information backing this statement.

*L. 595f: "…but surface bromine/halogen chemistry needs to be included to simulate springtime surface O3 at coastal Arctic locations …" I think it would be safe to say that halogen chemistry needs to be include to simulate springtime O3 in the entire Arctic and not only at coastal stations, where observations exist to compare the model output.*

[AR] Agreed. Reviewer 1 had a similar comment, and this sentence in the conclusion has been revised to account for the whole Arctic.

*Finally, there are some sensitive agreements between the current manuscript and the already mentioned paper by Whaley et al., 2022 (Atmos. Chem. Phys., 22, 5775–5828, 2022) when it comes to the model descriptions (see pages 6, 7 and 9 of the iThenticate.com Similarity Report). I leave it to the editor to decide if these matches need a revision.*

[AR] Thank you for mentioning this. This paper's model component is an expansion of previous work in Whaley et al (2022) in the sense that the models and their simulations are the same in this study as in that other study. We have added text in the revised manuscript, near the beginning of the model description section to be clear about that. Perhaps the editor can advise on whether re-writing/re-wording is necessary in the model description section.